# Counting cells can accurately predict small-molecule bioactivity benchmarks

Srijit Seal [1,2,11] ✉, William Dee [3,11], Adit Shah [2], Natacha Cerisier [4], Andrew Zhang [5], Esteban Miglietta [2], Katherine Titterton [6], Ángel Alexander Cabrera [6], Daniil Boiko [6], Alex Beatson [6], Gregory Slabaugh [3], Olivier Taboureau [4], Jordi Carreras Puigvert [7,8], Shantanu Singh [2], Ola Spjuth [7,8] ✉, Andreas Bender [1,9,10] ✉ & Anne E. Carpenter [2] ✉

Accurately predicting the activity of a chemical in each bioactivity assay based on its already known properties is extremely useful in drug development. Unfortunately, we discovered that many assays in widely used assay-activity benchmark datasets directly relate to cell health and cytotoxicity. Many other assays intend to capture a more specific phenotype, but their active compounds impact cell count, while inactives do not. In both cases, counting cells achieves unexpectedly high performance in these benchmarks, making them less useful for discerning whether additional properties, such as phenotypic profiles (mRNA or Cell Painting), provide additional useful information on bioactivity. To accomplish this goal, we recommend filtering benchmarks to exclude such assays and including a cell-count baseline. Using a benchmark with 24 protein-target assays, we confirm that models leveraging Cell Painting image-based profiles outperformed the baseline cell count model. We propose several other practical recommendations for benchmarking machine learning models for predicting bioactivity and assessing the added value of mRNA, protein, or image-based profiles.

Machine learning is now extensively used to predict the activity of chemical compounds in cell-based assays, classifying them as active, inactive, or inconclusive (or toxic, non-toxic, inconclusive)[1,2]. Such models can reduce the need for expensive laboratory experiments and potentially allow for predicting efficacy and toxicity early and inexpensively when developing drugs. Predictive models most commonly rely on chemical structures and features derived from them, such as molecular fingerprints, which are easy to generate[3,4]. However, improving model performance for compounds outside the training data's chemical space remains a critical challenge[5]. Capturing nuances such as activity cliffs, where similar structures show significantly

different bioactivity, has proven difficult even for deep-learning architectures, suggesting that chemical representations alone are insufficient for predicting many interactions in complex biological systems[6]. Also, extrapolating in chemical space is not trivial, and here, incorporating biological descriptors has the potential to increase the applicability domain of a model.

Phenotypic profiling techniques, such as gene expression and morphological profiling, have emerged as promising additional data sources for each compound, which might augment molecular structures in improving chemical activity prediction. Cell Painting is an image-based profiling method that is high-throughput and cost-

[1]Department of Chemistry, University of Cambridge, Cambridge, UK. [2]Broad Institute of MIT and Harvard, Cambridge, MA, USA. [3]Digital Environment Research Institute (DERI), Queen Mary University of London, London, UK. [4]Université Paris Cité, INSERM U1133, CNRS UMR 8251, Paris, France. [5]Health Sciences and Technology, Harvard-MIT, Cambridge, MA, USA. [6]Axiom Bio, San Francisco, CA, USA. [7]Department of Pharmaceutical Biosciences and Science for Life Laboratory, Uppsala University, Uppsala, Sweden. [8]Pixl Bio AB, Uppsala, Sweden. [9]Center for Biotechnology, College of Medicine and Health Sciences, Khalifa University of Science and Technology, Abu Dhabi, UAE. [10]STAR-UBB Institute, Babeş-Bolyai University, Cluj-Napoca, Romania. [11]These authors contributed equally: Srijit Seal, William Dee. ✉e-mail: srijit@understanding.bio; ola.spjuth@uu.se; andreas.bender@ku.ac.ae; anne@broadinstitute.org

effective and, therefore, has significant potential[7,8]. Cell Painting datasets provide rich phenotypic profiles by applying a range of fluorescent stains, which capture the cell's biological response to various compounds across various cellular compartments and substructures[9]. This assay has been reported to yield improved performance in downstream tasks compared to chemical structures alone[8,9]. Since the first demonstration that an image-based assay could predict seemingly unrelated assay outputs[10], a trend has emerged: using biological response descriptors, such as transcriptomics data or Cell Painting readouts, independently or combined with chemical fingerprints, to enhance predictive models[8,11].

For example, Hofmarcher et al. showed that convolutional neural networks trained on Cell Painting features predicted the outputs of 32% of 209 biological assays with area under the curve (AUC) values exceeding 0.9[12]. Moshkov et al. evaluated chemical structures, Cell Painting images, and gene expression profiles (from the L1000 assay) across 270 assays, finding that each modality independently contributed valuable information[13]. Fusing the three data modalities improved AUC for 62% of assays (167 out of 270), compared with using chemical structures alone. Sanchez-Fernandez et al. introduced CLOOME, a contrastive learning framework enabling cross-modal querying of Cell Painting images with chemical structures[14]. They also assessed the transferability of learned representations using linear probing on 209 assay activity prediction tasks with pre-trained image embeddings from CLOOME, which did not use activity data during training. The model achieved an average AUC of 0.714, outperforming fully supervised methods. Ha et al. benchmarked learning models using Cell Painting profiles to predict outcomes of 201 assays, an updated version of the Hofmarcher dataset[15]. When predicting for assays not included in training the models, they found models performed well across the novel prediction tasks, achieving AUC values ranging from 0.70 to 0.87 as the dataset size increased, effectively generalizing to new assays.

However, when reviewing the assays most accurately predicted in these prior studies, we observed that many were linked to cytotoxicity or cell proliferation. There is a complex interplay between the endpoint one aims to measure (or model) in an assay and various confounding factors, which might lead to misleading performance of any predictive models trained on the data[16,17]. While cytotoxicity/viability are sometimes the desired assay endpoint, these assays are often run side-by-side with an actual assay of interest (in the same cell type, for example) to improve assay accuracy by identifying and excluding compounds exhibiting cytotoxic effects from progressing further in development. Combining results from the primary assay and the cytotoxicity viability screen enables identifying compounds with specific biological effects, minimizing false positives and off-target effects. For example, Esher et al. addressed the challenge of distinguishing specific from nonspecific (cytotoxicity-triggered) reporter gene activation in six Tox21 high-throughput screening assays[16]. "Cytotoxicity burst" refers to the activation of stress responses at compound concentrations near the cell death threshold, leading to nonspecific assay results. The authors proposed to compare assay activity to baseline toxicity—the nonspecific accumulation of chemicals in cell membranes that disrupt cellular function at critical concentrations. Even without a specific target, compounds can cause potent toxicity due to hydrophobicity, potentially skewing assay results. They found that 37–87% of active hits in the six Tox21 assays they analyzed were likely due to cytotoxicity burst, while only 2–14% were specific[16]. This is plausible given assay responses can arise from multiple mechanisms, not all representing compounds with the intended pharmacological action[17]. Hence, this illustrates that, either directly or indirectly, assay endpoints that one measures (or predicts) can be correlated with confounding factors (often cytotoxicity), and not considering the latter in both measurements and predictive models gives misleading results.

Our originally anecdotal finding that a large proportion of the well-predicted assays in prior benchmarks were cytotoxicity/viability related prompted us to test and discover that simple cell counts are surprisingly predictive in many of these benchmark assay tasks. If simple metrics like cell count are driving much of the predictive power, it would call into question the added value of more complex features, such as Cell Painting or gene expression profiles. After systematic evaluations and confirming the above findings, we identified three practical implications: (1) biased benchmark datasets have a high proportion of cell viability assays; a fairly easy-to-predict assay endpoint that should not be overemphasized in benchmarking studies, (2) many assays in benchmark datasets are specific to targets but the active chemicals in the dataset also impact cell viability and the actual readout is confounded by effects like cytotoxicity burst[16], and (3) there is an absence of baseline models trained only on cell counts that can demonstrate the comparative advantage of a model trained on more complex phenotypic profiles.

Establishing simple baseline cell count models is essential in machine learning to assess the value of more complex solutions, akin to "Occam's Razor" when choosing the simplest model to explain a given observation[18,19]. Such baselines can include basic molecular features like "atom counts", which have shown strong performance in a virtual screening setting despite lacking structural information[20], or mean or majority-class classifiers in regression and classification settings, respectively[1,21,22]. Previous studies have benchmarked models that make other kinds of predictions using gene expression data[23]. For example, Csendes et al. found that simple baselines outperformed state-of-the-art foundation models in predicting post-perturbation gene expression[24]. Similarly, SpaDo showed that in spatial transcriptomics, simple clustering based on spatial neighbor cell type proportions could achieve comparable or better results than complex graph neural network models[25]. Motivated by these findings, we focused extensively on benchmarking phenotypic Cell Painting screening using data in this study. In this study, we propose a baseline machine learning model using cell count to assign activity probabilities, as it proves surprisingly effective in many assays, particularly human tumor cell line growth inhibition assays. We recommend that future studies carefully select assay categories for benchmarks and establish baseline cell count models to realistically gauge performance improvements when using high-dimensional phenotypic data, such as images, mRNA, or protein profiles, and present five related recommendations for the machine learning community.

## Results and discussion

### Multiple bioactivity tasks are highly correlated with the simple cell count feature

We first evaluated baseline cell count models trained on a single image-based feature, namely cell count based on segmentation of nuclei in a DNA-stained image, across four previously published bioactivity prediction benchmarks: (a) Moshkov et al.[13] (b) Hofmarcher et al.[12] (c) Sanchez-Fernandez et al. (CLOOME)[14], and (d) Ha et al.[15] (Table 1). This allows us to systematically assess the predictive utility of cell count as a prediction score to understand its predictive power. The Moshkov dataset, containing assay readouts generated internally at the Broad Institute, is distinct from the Hofmarcher and Ha datasets, both derived from ChEMBL. Sanchez-Fernandez et al. based their benchmarking on the Hofmarcher dataset. Although Hofmarcher and Ha originate from the same source, they were compiled at different times, leading to minor differences in compound coverage. While the assays vary slightly, all three datasets were evaluated against the same Cell Painting imaging dataset. To allow direct comparison with prior studies, we evaluated each dataset separately rather than combining them into a single benchmark.

We began by assessing the relationship between bioactivity prediction tasks and changes in cell count by comparing cell count

**Table 1 | Description of commonly used datasets compiling publicly available annotations for compounds in Cell Painting**

| Dataset | Number of compounds | Number of bioactivity endpoints ("tasks") | Type of tasks | Completeness | Percentage of active data points |
|---|---|---|---|---|---|
| Moshkov et al.[13] | 16,170 | 270 | Mostly cell-based compound activity and toxicity | 13.4% | 2.7% |
| Hofmarcher et al.[12] (also used in Sanchez et al.) | 10,573 | 209 | Protein target/bioactivity and tumor cell line growth inhibition assays | 2.5% | 34.7% |
| Ha et al.[15] (subset of Hofmarcher et al.[12]) | 10,526 | 201 | Protein target/bioactivity and tumor cell line growth inhibition assays | 2.6% | 34.7% |
| This Study (Eve Bio[66]) | 269 | 24 | Protein target/bioactivity | 91.6% | 13.71% |

Completeness: the percentage of compound-task pairings that have measured activity data; the percentage of active data points reflects the fraction of compound-assay pairings marked as active (bioactive, toxic, etc.).

distributions for active and inactive compounds across three benchmark datasets. We compared the mean cell count values of active versus inactive compounds, aggregated per assay, using a two-sided independent-samples t-test and found significant differences in all three benchmark datasets ($p < 10e\text{-}16$, Fig. 1a–c) suggesting that cell count alone might contain a substantial predictive signal for many biological assays.

We further analyzed the predictivity of the cell count feature for benchmark datasets used in prior studies and found that solely applying a range of thresholds of cell counts successfully predicted 32% of assays (66 out of 209) in the Hofmarcher dataset, with mean balanced accuracy over 0.70 (Fig. 1d), with 64 of the 66 assays being human tumor cell line growth inhibition assays. Thresholds of cell counts predicted 22% of the tasks in the Ha dataset and 28% in the Moshkov dataset (Fig. 1e, f) with mean balanced accuracy over 0.70. This indicates that these datasets contain a substantial number of assays whose outcomes primarily correlate with cell count and, therefore, are not ideally suited for assessing data modality or model effectiveness across a broad spectrum of assays.

Throughout this work, "baseline cell count model" refers to models using only the cell count feature, with the model type matched to each reanalyzed study for direct comparison: logistic regression for Moshkov et al. and Sanchez-Fernandez et al., and XGBoost for Hofmarcher et al. and Ha et al. These trained models learn the direction of association from data, unlike the fixed threshold analysis above (Fig. 1d–f, Eq. 1), which assumes low cell count indicates activity and requires manual output reversal when assay label conventions differ. To better contextualize the scope of what cell count captures, we analyzed whether the predictive signal from cell count correlates with cytotoxicity. Specifically, we combined the Moshkov and Hofmarcher datasets and stratified compounds by the number of assays in which they were labeled active. We compared the cell count values for compounds active in multiple assays and found that compounds labeled active in a greater number of these benchmark assays tend to show a stronger and more consistent decrease in cell count (Fig. 1g). While cell count is informative for strong and promiscuous cytotoxic responses, it is considerably less responsive to mechanism-specific or low-grade phenotypic changes, which may constrain its utility for detecting nuanced biological effects.

**Cell count matches the performance of models trained on Cell Painting and gene expression profiles on existing benchmarks**
We next investigated the information content in the cell count feature; for this, we followed the same train-test splits for each of the studies reanalyzed in this work, and compared our baseline cell count model performance to those models trained on the higher dimensionality data of Cell Painting and gene expression profiles.

In the Hofmarcher dataset[12], which was also analyzed in Sanchez-Fernandez et al.[14] and of which the Ha dataset[15] is a variation, the baseline cell count models (mean AUC = 0.68 ± 0.21) achieved similar performance compared to the supervised fully-connected neural network (FNN) model based on the full Cell Painting profiles (mean AUC = 0.67 ± 0.19), while the convolutional neural network architectures of ResNet model trained on Cell Painting images directly (mean AUC = 0.73 ± 0.19) performed better (Table 2, rows e, f, g). We observed a significant overlap of assays well-predicted across all three models (66 were well-predicted, AUC > 0.8), with the two more sophisticated image-based models collectively predicting only 19 additional assays over the baseline cell count model (Fig. 2a). We found performance of all models, whether using cell count or Cell Painting features, was largely based on human tumor cell line growth inhibition assays (Fig. 2b). Ideally, benchmarks should not over-represent a single type of assay, especially viability assays (Supplementary Fig. S1 and Supplementary Data 1), given that it is a readout that can be well-predicted by a single feature such as cell count (albeit with differences due to assay conditions such as cell type and timepoint).

The Hofmarcher dataset contains cell-based assays as well as biochemical binding assays using purified protein targets. Out of the 66 unique protein targets in the Hofmarcher dataset, cell count predicted only one protein target at a performance level of AUC > 0.8, while the convolutional neural network architectures of ResNet and DenseNet using images predict six targets and eleven targets respectively (Fig. 2b). This shows that while most methods perform similarly to baseline cell count models when evaluated on all assay types, when predicting small molecule activity on protein targets, Cell Painting images achieve high predictivity for many unique protein targets (for example, ATAD5, IDH1, ATXN2, MAPT, SMAD3, Hsf1, VDR, RAPGEF3, BRCA1, TDP1, and GMNN for convolutional neural network DenseNet model), many of which relate to DNA damage. This shows that the Cell Painting images are able to predict assay outcomes from many unique protein targets where the baseline cell count model fails.

We next investigated whether this substantial overlap in predictable assays was only feasible because of Cell Painting images' inherent ability to count cells or whether other unbiased profiling methods, such as mRNA profiling, might capture cytotoxicity/viability equally well. Using the Moshkov dataset[13], we found a substantial intersection between assays predicted by the baseline cell count model and Moshov's multi-task neural network trained on gene expression data (Fig. 2c), a similar result as for models using Cell Painting data. 31 out of 49 (63%) assay endpoints well-predicted using gene expression profiles (AUC > 0.7, threshold as used by the authors of the study, see Table 3d) were predicted equally well (AUC > 0.7) by using the baseline cell count model. Only five assays were uniquely well-predicted by gene expression profiles, while in contrast, 33 assays were uniquely well-predicted by incorporating the entire Cell Painting feature set and 22 assays were uniquely predicted by the baseline cell count model. Overall, we found that the performance of the baseline cell count models using a single cell count feature is comparable to models trained using Cell Painting and gene expression data for the assays evaluated in prior studies.

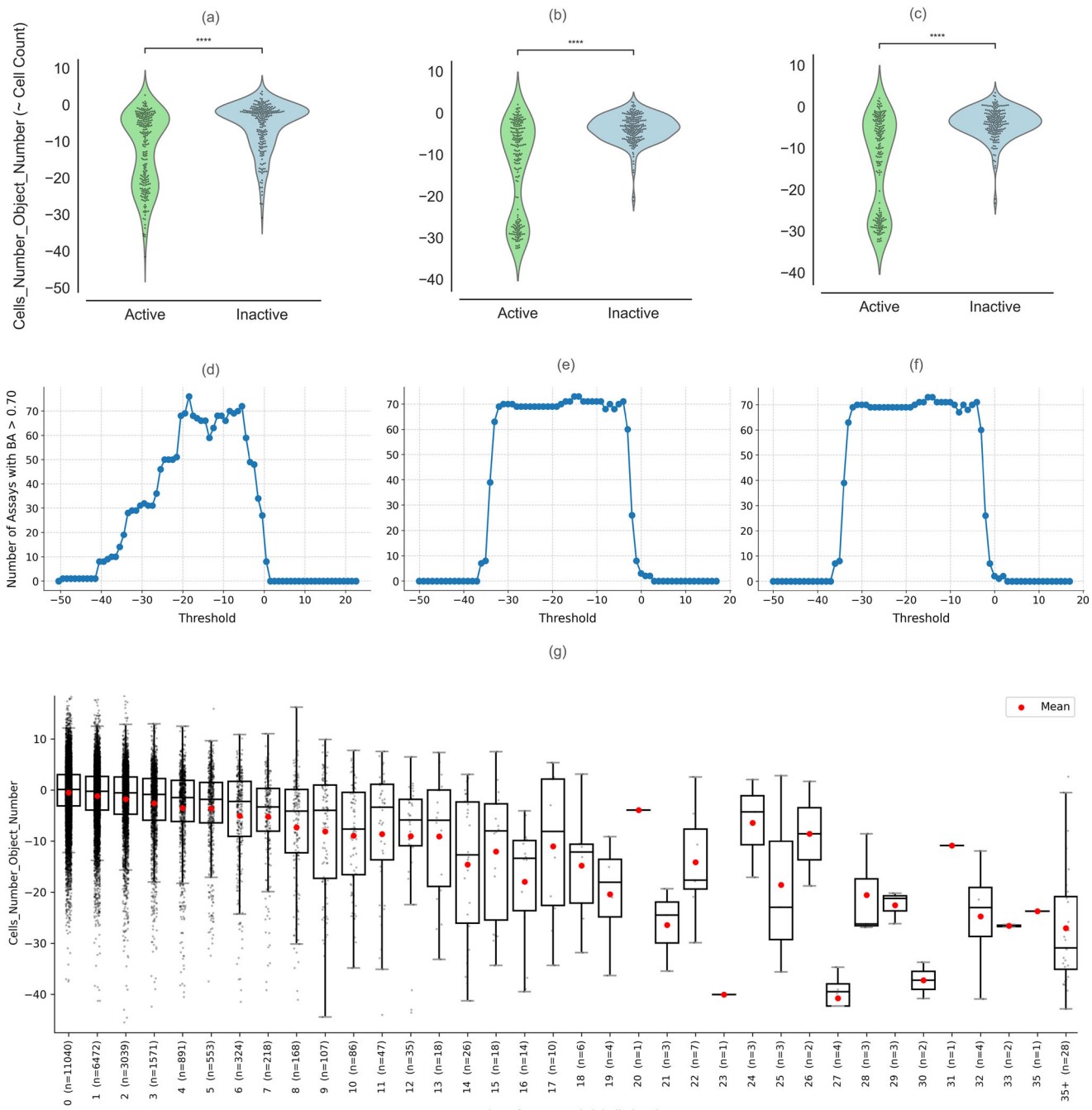

**Fig. 1 | Comparison of the distribution of the cell count feature ("Cells_Number_Object_Number") across.** **a** 270 assays in the Moshkov et al.[13] ($p < 2.2$e-16 stat = −8.572e+00), **b** 209 assays in the Hofmarcher et al.[12] ($p < 2.2$e-16 stat = −1.166e+01; which was also analyzed in Sanchez-Fernandez et al.[14]), and **c** 201 assays in the Ha et al.[15] ($p < 2.2$e-16 stat = −1.156e+01), split by active and inactive compounds and balanced to the same number of active and inactive compounds per assay. *P* values were obtained using a two-sided independent-samples t-test; values beyond machine precision are reported as $p < 2.2$e-16. Number of assays with balanced accuracies >0.70 when predicting assay outcomes in the **d** Moshkov dataset, **e** Hofmarcher dataset, and the **f** Ha dataset, by thresholding the normalized cell count feature at various cutoffs. **g** Relationship between compound promiscuity and cell count deviation. Compounds were grouped based on the number of assays in which they were active, using data from both the Moshkov and Hofmarcher datasets. Boxes represent the interquartile range (IQR; 25th–75th percentile), the horizontal line indicates the median, and whiskers extend to 1.5× IQR. Individual points represent single compounds. Red dots indicate the mean cell count. Error bars represent mean ± SD.

This analysis highlights the value of using cell counting models as an initial, interpretable approach before employing more complex features and to provide a baseline for later models.

## Gene enrichment for cytotoxic and promiscuous compounds

We hypothesized that (1) a decrease in cell number (Cells_Number_Object_Number) observed in Cell Painting assays may reflect increased apoptotic activity, and (2) compound promiscuity may arise more from general cytotoxicity (e.g., apoptosis) than from selective activity. To investigate this, we leveraged gene expression data from the LINCS L1000 resource to examine the biological processes associated with compound perturbations. For this analysis, we selected only those compounds from LINCS that passed the "is_exemplar" (which identifies a single "exemplar" signature for each perturbagen in each cell

**Table 2 | Performance metrics across 209 prediction tasks in the Hofmarcher dataset**

| Row | Type | Method | AUC | AUC > 0.9 | AUC > 0.8 | AUC > 0.7 | AUC > 0.5 |
|---|---|---|---|---|---|---|---|
| a | Linear probing | Logistic regression (CLOOME) | 0.71 ± 0.19 | 57 | 83 | 102 | 179 |
| b | Linear probing | Logistic regression (CellProfiler) | 0.66 ± 0.20 | 35 | 63 | 84 | 170 |
| c | Linear probing baseline (this study) | Logistic regression (Cell count) | 0.70 ± 0.22 | 64 | 74 | 103 | 189 |
| d | Linear probing baseline (this study) | Logistic regression (Cell count + MW + logP) | 0.74 ± 0.21 | 64 | 103 | 128 | 186 |
| e | Supervised | ResNet | 0.73 ± 0.19 | 66 | 85 | 114 | 178 |
| f | Supervised | FNN | 0.68 ± 0.20 | 54 | 69 | 78 | 170 |
| g | Supervised baseline (this study) | Cell count | 0.68 ± 0.21 | 57 | 73 | 82 | 168 |
| h | Supervised | DenseNet | 0.73 ± 0.19 | 65 | 93 | 113 | 178 |
| i | Supervised | GapNet | 0.73 ± 0.19 | 60 | 88 | 115 | 179 |
| j | Supervised | MIL-Net | 0.71 ± 0.18 | 59 | 71 | 100 | 187 |
| k | Supervised | M-CNN | 0.71 ± 0.19 | 58 | 72 | 100 | 172 |
| l | Supervised | SC-CNN | 0.71 ± 0.20 | 63 | 71 | 104 | 172 |

AUC values are reported as mean ± s.d. across the 209 assays where each assay's AUC is first averaged over three 70-10-20 train-test splits. These baseline cell count models are compared to models used in Hofmarcher et al.[12] and Sanchez-Fernandez et al.[14] (based on the complete Cell Painting features). Bold and underlined values indicate best and second-best performance.

line) quality filter and were tested in U2OS, MCF7, or PC3 cell lines. Of the 1,520 compounds that significantly reduced cell count in the Broad Cell Painting assay, 466 had corresponding LINCS mRNA signatures, yielding a total of 550 gene expression profiles.

Gene Ontology (GO) enrichment analysis of genes with a z-score threshold ($|z| > 2$) revealed that transcriptional changes that were induced by compounds known to reduce cell count (by Cell Painting) were consistently associated with particular biologically relevant processes. The most frequently enriched pathways included the intrinsic apoptotic signaling pathway ($n = 112$ compounds), mitotic cell cycle phase transitions ($n = 101$), regulation of cyclin-dependent kinase activity, spindle organization, and nuclear division. Stress response pathways (e.g., reactive oxygen species, hypoxia) and metabolic functions (e.g., cholesterol/sterol biosynthesis) were also enriched.

Analyzing more carefully the degree of cell count impact, we found that compounds more strongly reducing cell count in Cell Painting assays showed the highest enrichment for apoptosis- and cell death–related GO terms (Fig. 2e). Over 70% of compounds in the lowest cell count bins ([−37; −50], normalized feature values against plate-specific DMSO controls) were linked to apoptosis-related terms, with this proportion decreasing as cell count increased. This underscores the importance of cell health and cytotoxicity as major drivers of both morphological and transcriptomic profiles, and highlights the need to account for cell count when benchmarking both phenotypic profiling methods.

We next explored relationships between gene expression and compound promiscuity (activity across many assays). From the set of 24,606 compounds with target activity labels (from Hofmarcher and Moskhov datasets), we extracted 8381 LINCS signatures for 4094 compounds that passed the cell line and exemplar filters. Again, the intrinsic apoptotic signaling pathway was the most frequently enriched category ($n = 193$), supporting our hypothesis that compound promiscuity is often associated with general cytotoxic effects (Fig. 2f).

Beyond apoptosis, other recurrently enriched pathways associated with compound promiscuity included metabolic and biosynthetic processes (e.g., cholesterol/sterol biosynthesis [$n = 164–156$], secondary alcohol metabolism [$n = 164–124$], steroid biosynthesis [$n = 112$]), protein phosphorylation and kinase regulation (e.g., serine/threonine kinase activity [$n = 161$], protein kinase complex [$n = 131$]), and cell adhesion–related categories (e.g., cadherin binding [$n = 116$],

focal adhesion [$n = 89$]). Stress-related responses to ROS ($n = 101$), xenobiotics ($n = 86$), and hypoxia ($n = 84$) were also commonly enriched, highlighting the broad cellular impact of promiscuous compounds.

Focusing on the 214 compounds that reduced cell count and were also promiscuous (Fig. 2g), we found that apoptotic enrichment increased markedly as cell count decreased. In contrast, the effect of promiscuity on apoptosis enrichment was more modest (Fig. 2h). Notably, compounds with more than one active hit and a normalized cell count below −40 showed over 70% enrichment for apoptosis-related GO terms.

**Baseline cell count models show comparable AUC to meta-learning approaches**

We next compared our baseline cell count models with the single-task, multi-task, and meta-learning models using Cell Painting data used by Ha et al.[15]. We found that the baseline cell count model achieved a median AUC of 0.59 on this benchmark, whereas the "protonetcp+" model, a metric-based meta-learning model using embeddings from Cell Painting features, attained an AUC of 0.64 (Supplementary Data 2). Notably, with regard to the exceptional performance on specific assays, such as CHEMBL2114784[26] (AUC of 0.87), a viability screen for the primary quantitative high-throughput screening (qHTS) aimed at identifying inhibitors of ATXN expression, the baseline cell count model achieved an AUC of 0.82. The similarity in AUC scores suggests that the baseline is competitive with the more complex model, likely because the assay is linked to cell viability.

Furthermore, Ha et al. posited that pre-training significantly enhanced model performance on other assays, such as CHEMBL2114807[15,27]. On further examination, we found this qHTS assay was designed to identify small molecule activators of BRCA1 expression. The labeling for this assay was as follows: compounds with lower pChEMBL values were classified as activators and labeled "1". In comparison, those with higher pChEMBL values were labeled "0", representing likely inhibitors that could impair DNA repair mechanisms and induce cell death[27]. By reversing the output of the baseline cell count model to predict low cell counts as "0", we achieved an AUC of 0.76, closely matching the best pre-trained models reported by the authors (AUC = 0.77). Hence, we can conclude that it is key to use data in a suitable way when training models and also that cell counts are very predictive for this particular endpoint.

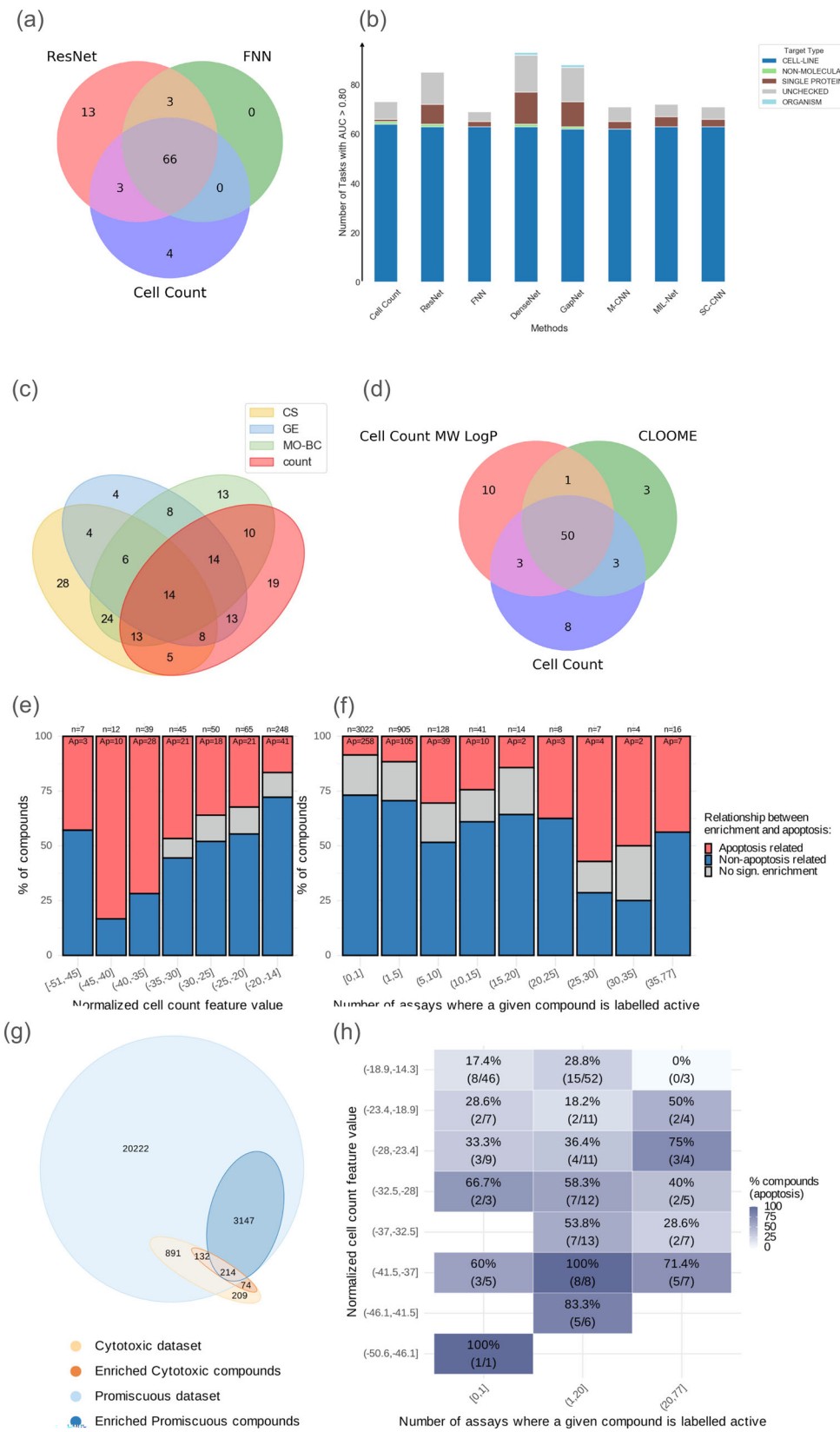

These results underscore a cautionary reminder regarding interpreting AUC values: models predicting assays related to cytotoxicity cannot be easily compared with each other; often, in the case of meta-learning, test assays predict the opposite outcome of the training assays and a model, being unaware of this, predicts the opposite of the

assay outcomes precisely. Given that the user always knows what assays it intends to predict, this task does not need meta-learning but merely reversing the prediction outcomes, as demonstrated here. Thus, AUC values provide only part of, but not the most important part of, performance[28].

**Fig. 2 | Performance of morphology- and structure-based models across diverse assay datasets and their association with cytotoxicity, apoptosis enrichment, and compound promiscuity. a** The number of assays and **b** assay types predicted with AUC > 0.8 in the Hofmarcher dataset, **c** the number of assays with AUC > 0.7 in the Moshkov dataset, and **d** number of assays with AUC > 0.9 with linear probing models using CLOOME image and chemical structure-based embeddings in Sanchez-Fernandez et al compared to baseline cell count models. Multi-task neural network (results from Moshkov et al.) were trained using either CellProfiler morphological profiles, which have been batch-corrected (MOBC), or gene expression data (GE). Baseline cell count models were trained only using the "Cells_Number_-Object_Number" morphology feature (cell count). Differing AUC cutoffs are used to be consistent with the originally published results for each dataset. **e** Proportion of compounds enriched for apoptosis/cell death (red), other GO terms (blue), or no significant GO terms (gray), stratified by normalized cell counts bins, standardized against plate-specific DMSO controls. Compound counts per bin are shown above

bars. Apoptosis-related enrichment exceeded 70% in two of the three lowest bins and declined with increasing cell count. **f** Similar analysis of compound promiscuity (i.e., number of active target labels for each compound). Apoptosis enrichment generally increases among compounds with an increase of active labels. **g** Euler diagram showing overlap between cytotoxic ($n = 1520$; orange) and promiscuous compounds ($n = 24,606$; blue; "promiscuous" here refers to compounds with available bioactivity annotations across the Hofmarcher and Moshkov datasets, not a minimum activity threshold), before (light) and after (dark) GO enrichment filtering. 214 compounds were shared between the two enriched subsets. **h** Heatmap showing apoptosis enrichment (%) across combinations of cell count bins and promiscuity bins. Each tile indicates the percentage and count (apoptosis / total). Color intensity reflects the level of apoptotic enrichment. FNN Fully connected neural network, ResNet residual neural network, CLOOME Contrastive Learning and leave-One-Out-boost for Molecule Encoders.

**Table 3 | Performance metrics across 230[a] prediction tasks in the Moshkov dataset when using a baseline cell count model, compared to late fusion models evaluated in Moshkov et al.[13]**

| Row | Method | Features | Mean AUC | AUC > 0.9 | AUC > 0.7 | AUC > 0.5 |
|---|---|---|---|---|---|---|
| a | Logistic regression baseline | Cell count | 0.611 ± 0.145 | 11 | 72 | 179 |
| b | Logistic regression | MOBC | 0.613 ± 0.156 | 13 | 83 | 179 |
| c | Multi-task neural network[13] | MOBC | 0.628 ± 0.170 | 9 | 84 | 189 |
| d | Multi-task neural network[13] | GE | 0.596 ± 0.140 | 5 | 49 | 180 |
| e | Multi-task neural network (late fusion)[13] | CS, MOBC | 0.649 ± 0.183 | 13 | 99 | 190 |
| f | Multi-task neural network (late fusion)[13] | GE, MOBC | 0.650 ± 0.149 | 10 | 89 | 196 |
| g | Multi-task neural network (late fusion)[13] | CS, GE, MOBC | 0.658 ± 0.159 | 10 | 94 | 195 |
| h | y-scrambling control | MOBC | 0.483 ± 0.207 | 1 | 15 | 97 |

Bold and underlined values indicate best and second-best performance.

*CS* chemical structure, *GE* gene expression, *MOBC* morphology, batch corrected - Cell Painting.

[a]40 assays have been excluded, which did not contain enough data points to guarantee that at least one active and one inactive compound is included in the training and test data for each fold in cross-validation. Therefore, results from Moshkov et al.[13] have been updated to exclude these assays.

## Contrastively learned features do not outperform cell count features

We next evaluated whether contrastive-learned morphological features learn more than just cell count, in the context of assay prediction. We used features learned using CLOOME (a contrastive learning framework that pairs microscopy images and chemical structures to enhance molecular representation learning) to two logistic regression models: one trained with the cell count feature and another with cell count, and two readily-calculated features of chemical structure: molecular weight and logP[14]. On the Hofmarcher dataset and the publicly available CLOOME embeddings, we tested linear probing, which is a logistic regression task commonly used to assess the quality of representations learned through contrastive learning. It achieved an average AUC of 0.71 ± 0.19 and 57 well-predicted assays compared to using the cell count feature, which achieved a similar average AUC of 0.70 ± 0.22 and 64 well-predicted assays (Table 2, rows a and b). We found that furthermore the logistic regression model trained on cell count, molecular weight, and logP values achieved an average AUC (0.74 ± 0.21) and 64 assays with AUC > 0.9, thus even slightly superior to CLOOME (Fig. 2d). Overall, this highlights the importance of cell counting as a baseline cell count model and the choice of a suitable task/endpoint for model benchmarking, given there was no observed improvement in performance when using contrastive learning embedding.

## Appropriate evaluation metrics and assay endpoints are essential when modeling large-scale benchmark datasets

The multi-task neural network models trained on complete Cell Painting profiles in Moshkov et al. achieved extremely high AUC values

(1.0) for nine out of the original 270 assays[13]. Eight of these endpoints contained only one active compound, while one contained two actives. Each endpoint also contained fewer than 52 data points in total. This lack of data, combined with the scarcity of active compounds, made high AUCs achievable by chance, by solely predicting one active compound correctly. On inspection, nine of the ten unique compounds labeled as active for these assays were cytotoxic[29–31], leading to significant cell death and low cell density in images. Example images of cells, treated with four different compounds (fluspirilene, hinokitiol, pyrazolanthrone, BRD-A64553394) and neutral control DMSO show significant cell death, dark areas, and low cell density, indicating cytotoxicity (the most extreme cases are bordered yellow) are shown in Fig. 3. The images (Fig. 3a–d) may not have been used in generating profiles, as a QC pipeline in CellProfiler is likely to reject them and only keep some images with some surviving cells[32]; nevertheless, the images show that these compounds are cytotoxic. Therefore, the high performance could also be attributed to models effectively predicting cytotoxicity (low cell counts).

This finding is supported by comparing the AUC scores of the different model approaches after separating the remaining 230 assay endpoints into two groups: 185 assays containing five or more active compounds and assays containing fewer than five active compounds (Supplementary Fig. S2). This reveals that a greater proportion of endpoints are predicted with higher AUC when there are fewer active compounds in the data. This is particularly true in the case of our baseline cell count model, where the use of one feature and a simpler model architecture likely led to less overfitting. Conversely, as the size of the training data increases, a baseline cell count model underperforms the two approaches trained using all CellProfiler features. This

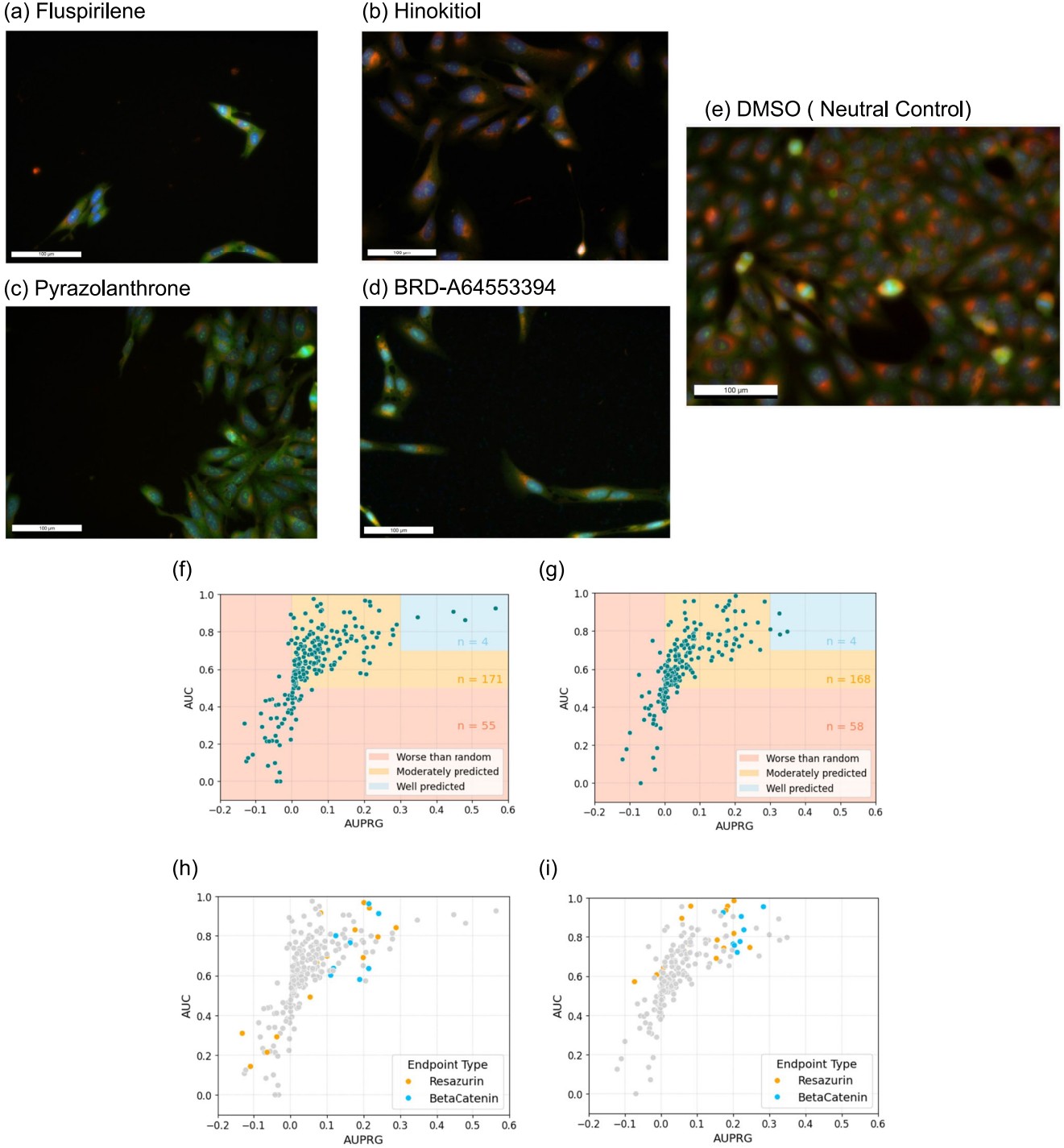

**Fig. 3 | Example images of cells treated with four different compounds.**
**a** fluspirilene, **b** hinokitiol, **c** pyrazolanthrone, and **d** BRD-A64553394, along with **e** the neutral control DMSO. The treated wells show marked cytotoxicity, characterized by extensive cell death, dark regions, and low cell density (extreme cases highlighted in yellow). Although these images (**a**–**d**) may not have contributed to profile generation because CellProfiler's QC pipeline likely filters them out in favor of images containing some surviving cells, they clearly demonstrate the cytotoxic nature of these compounds. Representative images were selected from $n = 48$, 48, 48, 42, and 16,174 independent fields of view for fluspirilene, hinokitiol, pyrazolanthrone, BRD-A64553394, and DMSO, respectively. Images from additional sites and replicates for these compounds show consistent patterns and can be viewed at https://idr.openmicroscopy.org. Comparison of the performance of logistic regression models incorporating either **f** all CellProfiler features or **g** the cell count baseline model using only the "Cells_Number_Object_Number" feature. Area under the receiver operating curve (AUC) has been plotted against area under the precision-recall-gain curve (AURPG) to appropriately compare performance in the Moshkov dataset, where there is significant data imbalance. Specific endpoints of interest have been highlighted relating to either Resazurin (orange) or BetaCatenin (blue) assays for the logistic regression models trained with **h** all CellProfiler features or **i** only the cell count feature. These two assays typically identify active compounds that cause reduced cell viability, and it is shown above that in these instances, the cell count baseline model demonstrates strong predictive ability.

demonstrates that, for appropriate endpoints, more complex approaches are deriving additional value from high-dimensional feature sets beyond cell count.

To investigate whether the sole use of AUC could be misleading in terms of judging model effectiveness, for the remaining 230 assays in the Moshkov dataset, we plotted the AUC of each model against AUPRG, which, as mentioned above, provides a universal baseline for comparison between assays that is less impacted by severe class imbalance (Fig. 3f, g). Thresholds to categorize assay endpoints as "well-predicted" were set at 0.7 AUC and 0.3 AUPRG. Using the more stringent performance criteria meant that just 4 assays were well-predicted by our logistic regression model (Fig. 3f), in contrast to the 83 assays predicted with AUC > 0.7 or 13 with AUC > 0.9 (Table 3b). This reduction shows that AUC alone was insufficient to assess the number of "well-predicted" assays.

We also hypothesized that a number of the assay endpoints were highly correlated with cell death and thus would be predicted with high performance by a baseline cell count model. We therefore focused on two distinct endpoint types within the data: resazurin and Wnt/β-catenin signaling pathway assays. The resazurin assay is commonly used to assess both bacterial and eukaryotic cell viability[33,34], whilst the Wnt/β-catenin signaling pathway is crucial for cell proliferation and survival[35]. The Cell Painting experiments used U2OS osteosarcoma cells where Wnt signaling is often active[36,37]; therefore, it is likely that active inhibitory compounds would result in reduced cell proliferation, as well as increased apoptosis and cell cycle disruption. These endpoint types accounted for 14 and eight assays, respectively, out of the 230 assays in the data. Generally, these two endpoint types were well-predicted by our baseline cell count model, with an average 0.806 AUC and 0.154 AURPG, outperforming the 0.70 average AUC achieved by Moshkov et al.'s late fusion of three different data modalities for those endpoints. Only two out of 22 assays (9%) were predicted worse than random by the baseline cell count model (Fig. 3i). However, the logistic regression model trained on all CellProfiler features results in worse model performance on these endpoint types (Fig. 3h), with five resazurin assay endpoints predicted with AUC < 0.5. Instead, other assays are predicted with greater AUC/AURPG scores in their place, resulting in a similar number of assays above our determined performance thresholds (see Tables 2a and 3b). This finding supports the necessity to filter these benchmark datasets, to focus on endpoints where applying advanced deep learning techniques will yield real benefits over and above the information contained in cell count.

To assess whether the cell count baseline and the full Cell Painting feature model were identifying the same compounds as active, we examined the true positive prediction agreement between the two models across all assay endpoints in the Moshkov dataset. There was a 49.54% overlap between the true positive predictions, suggesting the approaches are partially predicting different groups of compounds as active. To investigate the difference in these compound cohorts, we isolated the compounds that were uniquely predicted by each model. We then calculated the following physicochemical properties for each compound molecule: molecular weight (MW), lipophilicity (LogP), topological polar surface area (TPSA), hydrogen bond donors (HBD) and acceptors (HBA) and rotatable bonds. We also computed the statistical significance of the difference of the means of each metric between the uniquely predicted compounds for each model using scipy's "ttest_ind" ($p < 0.05$).

The true positive compounds predicted uniquely by the model trained only using cell count were found to have statistically significantly lower average MW and LogP, while having a higher average TPSA and more rotatable bonds when compared with the cohort of compounds uniquely predicted by the model trained using all CellProfiler features from Cell Painting (Supplementary Fig. S3). Generally, lower LogP combined with higher TPSA indicates that the compound should exhibit better aqueous solubility[38,39]. Lower MW compounds also tend to diffuse more easily across cell membranes, and more rotatable bonds indicate higher molecular flexibility— more promiscuous binding and greater likelihood of fitting into various binding sites[40]. Taken in combination, this demonstrates that the cell count model is more adept at identifying compounds with more broad, simplistic effects and higher general/nonspecific cytotoxicity as they are more likely to impact broad cellular processes. Conversely, the model trained on all features can categorize compounds that have a lower likelihood of being reliably delivered to cells or exhibit more subtle or specific phenotypes.

To further explore these findings, we examined the structural diversity of compounds that were correctly predicted as active by only one of the models. When comparing compound scaffolds, we observed substantially lower scaffold diversity among the compounds uniquely identified by the cell count model (528 unique scaffolds) compared with those uniquely identified by the full Cell Painting feature model (1175 unique scaffolds). This further supports the finding that the cell count model is more narrow in its predictive ability, but when applied in a setting where a high number of compounds are cytotoxic, or assays are primarily aimed at measuring cell viability, this approach can appear to perform on par with a model utilizing all CellProfiler features, demonstrating the necessity for the benchmark curation suggested within this paper.

## Consistent cell count reduction across plates shows that benchmark active compounds often induce cell death

Cell count can be impacted by many technical variables, raising the potential for plate or batch effects to confound analysis. In this study, we leveraged Cell Painting data normalized against plate-specific DMSO controls and aggregated across replicates to mitigate technical variability inherent to high-throughput imaging assays. We also evaluated the cell count feature (among others) before aggregation and found that, although artifacts exist that can affect cell count, the predictive power of cell counts is not an artifact, but rather a real phenotypic effect. While the assay outputs we aim to predict may themselves be subject to technical or batch effects, it is important to note that these readouts originate from independent experiments distinct from the imaging data, with different plate layouts and batching, and without marked grouping of activities on particular plates.

Specifically, we found that of 402 plates from the Broad Cell Painting dataset, 99% contained both bioactive and inactive compounds, defined by activity annotations from the Hofmarcher, Ha, or Sanchez datasets. Notably, across these plates, bioactive compounds exhibited a consistent reduction in mean cell count relative to inactive compounds in 88% of cases, even before aggregating features at the compound level (Fig. 4a). This reproducibility supports the relevance of cell count as a sensitive phenotypic marker.

We found that variability in pre-normalized cell count ranks moderately (94th out of 1783 features) in terms of mean standard deviation across plates. Features such as nuclei area shape exhibited even higher inter-plate variation (Fig. 4b). Overall, the consistent decrease in cell count observed across nearly all plates for bioactive compounds reinforces confidence in cell count as a phenotypic marker; however, this also highlights that the benchmark assays currently used are heavily biased toward viability-related endpoints, and consequently, the identified bioactive compounds largely reflect viability effects.

Next, we aggregated the Cell Painting features as medians across replicates, and we filtered compounds to retain unique compound skeletons in the bioactive and inactive groups, minimizing structural redundancy. Still, we found that bioactive compounds tend to have a larger number of Cell Painting features exhibiting higher absolute correlation with the cell count feature compared to inactive compounds (Fig. 4c).

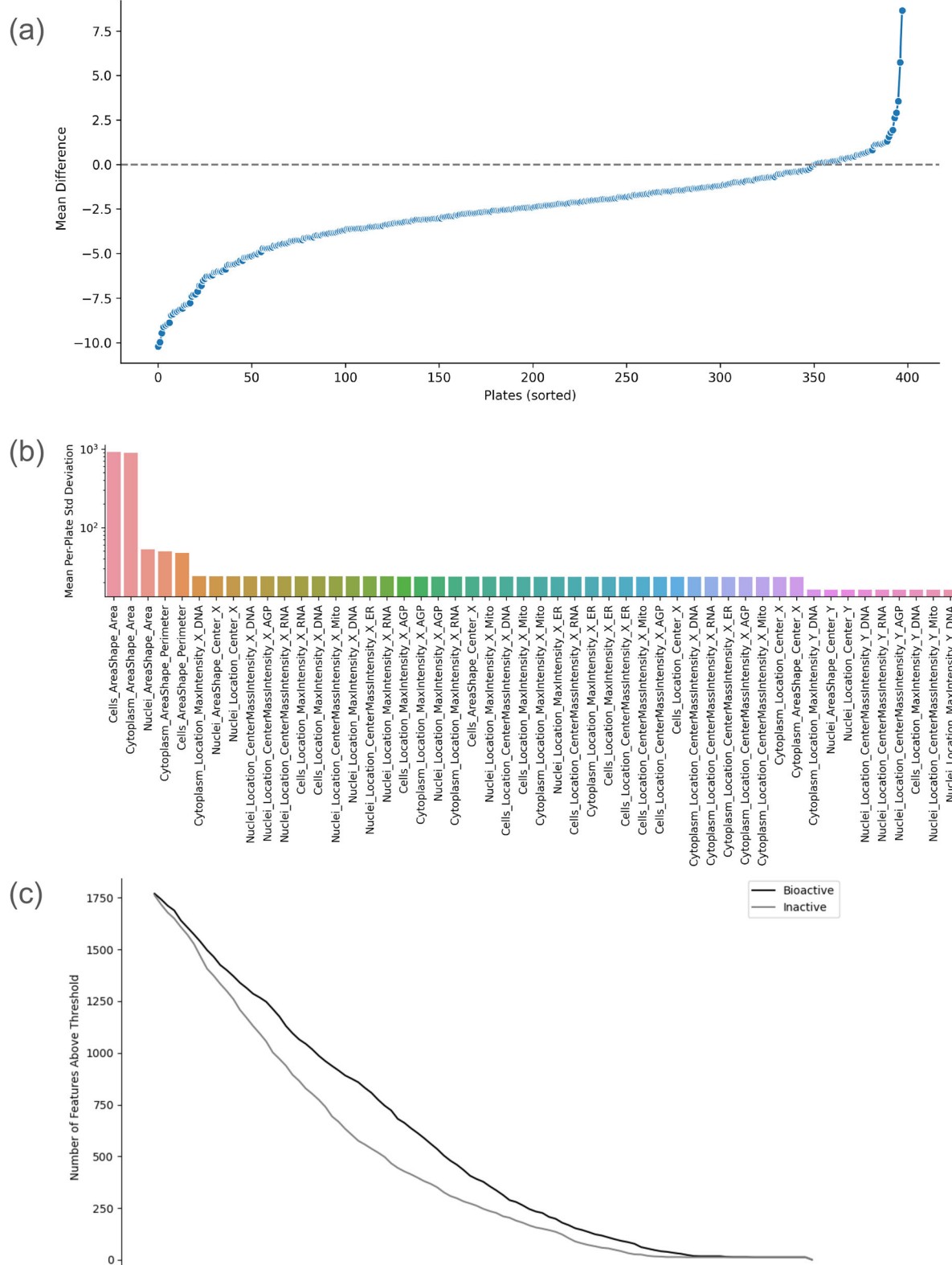

**Fig. 4 | Limited inter-plate variability and discriminatory power of cell count features relative to the full Cell Painting feature space. a** Distribution of the difference between mean cell count values for bioactive versus inactive compounds across 402 plates from the Broad Cell Painting dataset. **b** Ranking of top 50 Cell Painting features by mean standard deviation across plates; cell count (not shown) ranks 94th out of 1783 features in inter-plate variability. **c** Distribution of the number of features with high absolute correlation to the cell count feature (after aggregation) for bioactive and inactive compounds (after chemical skeleton level deduplication).

These findings suggest that bioactive compounds, likely those that reduce cell count, are associated with diverse phenotypic changes captured by the Cell Painting features. Overall, we found that the predictive power of cell counts is not an artifact of actives being concentrated on particular plates in the experiment, but rather a real phenotypic effect.

## Comparing compounds to positive controls is not a good benchmark for ML models

Cell Painting experiments use positive controls, which are typically compounds that exhibit a strong phenotypic change, often including a reduction in cell counts. Therefore, studies that compare methods by evaluating whether models can predict if a compound is similar to positive controls are likely confounded by cell count differences. For example, Cross-Zamirski et al. demonstrated label-free prediction of Cell Painting profiles from brightfield images[41]. This approach enabled the clustering of promising compounds with a positive control compound using a k-NN classifier applied to features from actual Cell Painting fluorescent images and predicted ones derived from brightfield images. The authors reported a sensitivity of 62.5% and a specificity of 99.3% when predicting cytotoxicity, with correct classifications for eight toxic compounds.

Among the label-free models, the feature most strongly correlated with the ground truth was Cells_Neighbors_SecondClosestObjectNumber_5 (Pearson correlation = 0.98), which is closely linked to cell count. Using this feature, we evaluated the 170 treatment wells in the test set and identified eight treatment compounds within 1.5 standard deviations of the positive control's feature value (Fig. 5a). When k-NN models were trained on either the full Cell Painting profiles (ground truth) or the single cell neighbor feature, they identified 7 and 19 treatment wells, respectively, as similar to the positive control. In contrast, label-free Cell Painting profiles generated using cWGAN-GP and U-Net models identified 10 and 14 wells, respectively. As shown in Fig. 5b, the majority of wells identified by the label-free models were encompassed within the predictions of the baseline cell count model. Because the dataset used by Cross-Zamirski et al. is not publicly shared, it remains unclear whether these well-level predictions align with their reported compound-level classifications. However, the strong correlation observed between the ground truth and the label-free cell count feature suggests that the predicted compounds from both datasets likely overlap. This evidence highlights the importance of including a baseline cell count model when presenting more complex models. Both Cell Painting and label-free models consistently identified compounds causing significant decreases in cell count, similar to the positive control compound mitoxantrone, a known cytotoxicant. Therefore, baseline cell count models serve as a reference point for evaluating and interpreting results from more complex models, particularly when the task is linked to separating positive controls from other perturbations.

## Concentration-dependent effects enhance information content in Cell Painting data

Most studies use data from two large public Cell Painting datasets: Bray et al.[42] and JUMP-CP[43]. These datasets primarily consist of single-dose experiments, where each compound perturbation was tested in multiple replicates but at one concentration, usually 10 μM. A single-dose setup makes it difficult to ascertain whether the compound concentration is optimal to capture unique effects of the compound, or alternatively, whether it is higher than desired and begins to capture effects associated with different mechanisms, including those that are outright cytotoxic (conditions where baseline cell count models perform well). Hence, we next explored whether the information content in Cell Painting profiles is concentration-dependent, using datasets with multiple compound concentrations.

To this end, we used data from a recent study by Comolet et al., which developed ScaleFEx, a memory-efficient and open-source pipeline for extracting interpretable cellular features from large high-content imaging datasets[44]. We evaluated the ScaleFEx feature space to identify phenotypic shifts in drug-treated cells. For example, the study validated ScaleFEx features on fibroblasts treated with a single drug, Y-39983-HCl, which revealed dose-dependent phenotypic changes. At 1.0 μM, the drug caused significant alterations, including reduced cell size and irregular cell shape, aligning with its known disruption of actin filaments. The authors showed that a logistic regression model trained in a five-fold cross-validation with plate held-out splits was able to differentiate the drug from the controls with high accuracy (AUC = 1.0). At a lower dose of 0.2 μM, more subtle but still significant differences were observed, such as decreased nuclear eccentricity and altered cell granularity; this shows that lower doses can still induce phenotypic changes that are useful. We note that a leave-one-plate-out cross validation, where unique perturbations (compounds and dose) are spread across different plates, might leak information in most modeling approaches; however, in this case, it is suitable for accounting for what makes a compound perturbation unique to the imaging across the experiment. Therefore, we retained the same leave-one-plate-out cross validation as the original authors.

We reproduced this experiment with only a single feature ("Mito-Count", the number of individual counts in the Mitochondria skeletonized mask, which is the closest related available feature to cell count) in order to compare their approach with a baseline cell count model. We found there was still a significant separation between the compound, Y-39983-HCl (1.0 μM), and controls when using the LR model with only MitoCount as a feature (AUC = 0.85), as shown in Fig. 5c–f. For other compounds (including Y-39983-HCl at 0.2 μM), we observed better performance for ScaleFEx features (mean AUC = 0.99) compared to baseline models using MitoCount (mean AUC = 0.66). While the original paper identifies Zernike_ch1_1 as one of the features contributing to model performance and demonstrates its ability to distinguish Y-39983-HCl (1.0 μM) from the control, we additionally show that the MitCount feature achieves a comparable level of significance in this separation (Fig. 5f). Overall, this shows that a baseline cell count-related model is a better benchmark than a random model. It also demonstrates that where phenotypic activity is subtle (at low concentrations), the full morphological profiles outperform a single feature related to cell count (as observed in the case of Y-39983-HCl at 0.2 μM dose).

To explore the concentration-dependency of information in Cell Painting profiles further, we evaluated the dose-response dataset as a proof of concept (an earlier version of this dataset is described in Ewald et al.[45]). We chose six compounds, chosen to highlight the separation between cytotoxic and bioactivity signals along concentration-response curves. Specifically, we chose three compounds that induced significant cell death at higher concentrations (Fig. 6a–c). We also selected three compounds that exhibited distinct phenotypes without cytotoxicity across the tested concentration range (Fig. 6d–f). Notably, the three cytotoxic compounds showed early morphological changes at concentrations ten times lower than those causing detectable cell death, as indicated by nucleus counts (Fig. 6a–c). Although these findings are preliminary and specific to the six compounds, this is consistent with our larger analysis in Ewald et al., which found that across >1000 compounds, early morphology changes were observed at doses about an order of magnitude lower than cytotoxicity readouts. Further studies affirm that for most compounds that exhibit morphological changes, the information content in Cell Painting is dependent on dose[45].

## Recommendations for model benchmarking

Given the above analyses, five practical recommendations are proposed for evaluating machine learning models using high-dimensional phenotypic profiles:

(1)    Benchmark datasets should be evaluated for assay correlations to avoid over-representation of cytotoxicity or other highly

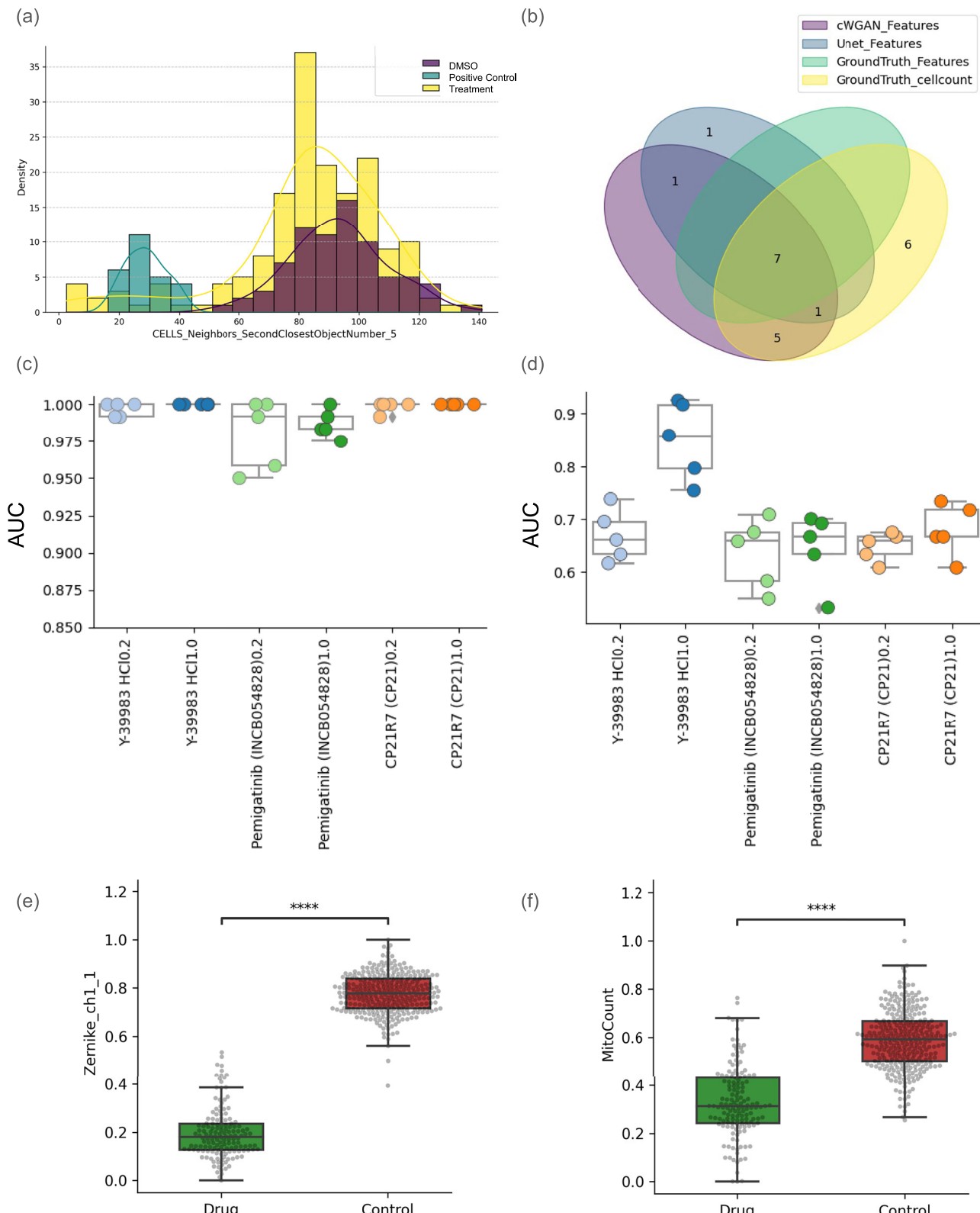

similar assays, which limit model generalizability. Selection of assay endpoints is critical for meaningful benchmarking of predictive models using Cell Painting or similar phenotypic profiling techniques. Current benchmarks often focus on viability-related assays like cytotoxicity or proliferation, where simple features (e.g., cell count) can outperform complex profiles, undermining the value of high-dimensional data. To improve evaluation, benchmarks should include diverse,

biologically nuanced endpoints (such as specific protein targets) where complex features provide real predictive advantage. Curating such balanced assay panels minimizes bias and enables more accurate model assessment.

(2) To avoid misinterpreting cytotoxic effects as pathway-specific activity, we recommend: (a) assessing the distribution of cytotoxic compounds per task using viability data or cell counts from Cell Painting (where available), and (b) reviewing images of

**Fig. 5 | Cell count−driven effects underlie apparent treatment similarity to positive controls across diverse modeling approaches. a** Distribution of the Cells_Neighbors_SecondClosestObjectNumber_5 feature for positive and negative controls, and treatment wells in Cross-Zamirski et al. It shows that positive controls have reduced cell count compared to DMSO controls and most other treatments. **b** Overlap of the wells identified as similar to the positive control by different approaches: the label-free models (cWGAN-GP and U-Net), the full Cell Painting profile and the baseline model using only the cell neighbor feature, which captures cell count indirectly. Distribution of AUC values for **c** the model used in Comolet et al. and **d** the baseline cell-count model using the MitoCount feature, evaluated for each drug versus its matched DMSO control using a linear regression model

with 5-fold cross-validation. Each point represents a different fold of the cross-validation. **e**, **f** show the distribution of the most important feature for correctly classifying Y-39983 HCl at 1.0 μM ($n = 148$) from StockDMSO ($n = 300$) for the Comolet et al. model ($U = 13.0$, $p < 2.2e\text{-}16$ and the baseline MitoCount model ($U = 4269.0$, $p < 2.2e\text{-}16$), respectively. Feature values are normalized to the range 0 to 1. Box plots display the median (center line), first and third quartiles (box edges), and whiskers extending to 1.5× the interquartile range; points outside this range are plotted individually as outliers. Statistical significance was assessed using a two-sided Mann−Whitney U test and is reported using the following thresholds: ns ($p > 0.05$), ∗ ($0.01 < p \le 0.05$), and ∗∗∗∗ ($p \le 0.0001$).

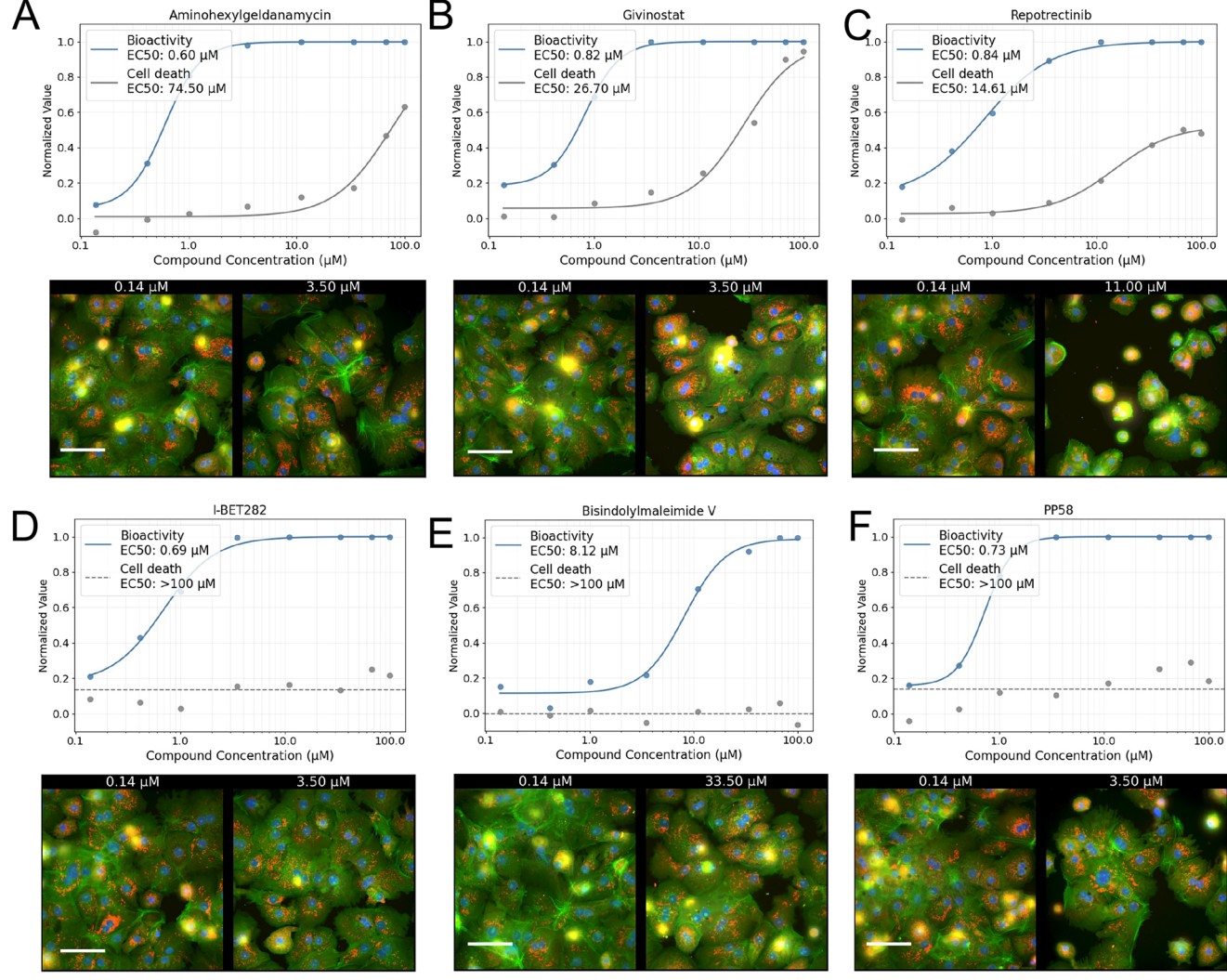

DAPI / WGP / Mitotracker

**Fig. 6 | Image-based bioactivity screen with a viability screen for cytotoxicity.** Bioactivity here refers to the morphological changes induced by compounds, measured as classifier probabilities distinguishing treated cells from DMSO controls. A probability of 0 indicates no detectable activity (indistinguishable from DMSO), while 1 represents significant morphological changes compared to DMSO. We show example dose-response curves for morphology change (bioactivity, blue), with higher values showing a perturbation being more dissimilar to DMSO, and cell death (nuclei count, gray), where higher values represent higher cell death as measured using nuclei count. **A**−**C** Example response curves for three cytotoxic compounds with various mechanisms of action (MOAs): **A** Aminohexylgeldanamycin, an Hsp90 inhibitor **B** Givinostat, a histone deacetylase

(HDAC) inhibitor, and **C** Repotrectinib, a tyrosine kinase inhibitor. Below each dose-response curve, representative 40× single field images for the lowest dose (0.14 μM) and first dose with very significant bioactivity (>0.9) are depicted. The scale bar is 80 μM. Cytotoxic compounds shrink the cytosol, and there are breaks in confluency. **D**, **E** Example response curves for compounds that were significantly bioactive but not did not considerably reduce nuclei count, including **D** I-BET282 pan-inhibitor of all eight BET bromodomains, **E** Bisindolylmaleimide V, protein kinase C inhibitor, and **F** PP58, Src Inhibitor. All dose-response experiments and image acquisitions were performed in two independent biological replicates. Representative visuals displayed here are taken from one replicate.

confident, correct predictions[16]. Assays with high proportions of cytotoxic compounds should be flagged or filtered. It's also critical to verify that the predictive target assay reflects specific compound activity and not viability data mislabeled in sources like PubChem. For Cell Painting datasets, apply QC post-processing, plate/batch corrections, and remove compounds with severely reduced cell counts (visually or by metrics). Features strongly correlated with cell count (e.g., $|r| > 0.95$) should be excluded to reduce confounding. Aggregate replicates (mean/median) only after correcting artifacts like plate-row effects. After modeling, inspect raw images of confident predictions for signs of cytotoxicity (e.g., few cells). The images from Bray et al. (https://idr.openmicroscopy.org/search) and the JUMP-CP project (https://phenaid.ardigen.com/jumpcpexplorer) are publicly searchable, providing resources to facilitate such examinations[42,43]. For transcriptomic/proteomic data, interpret key predictive features to ensure they reflect pathway-specific biology rather than general stress or death responses.

(3) Include a cell count model as a baseline. This offers a valuable benchmark for performance metrics, especially when a dataset contains assays directly linked to cell viability and/or when active compounds are disproportionately cytotoxic. The baseline cell count model can be a logistic regression model trained on the cell count feature (or a related/correlated feature like cell neighbor count, mitocount, etc.), but otherwise keeping the training/evaluation approach identical. This will reveal whether more complex models capture insights beyond cytotoxicity.

(4) Ensure a sufficient number and diversity of active compounds in the test sets. An insufficient number of test compounds, particularly with imbalanced classes, can lead to inflated performance metrics. For example, correctly predicting a single active compound in a test fold could misleadingly result in an AUC of 1. To mitigate this, Hofmarcher et al.[12] ensured that each assay in their dataset included at least ten active and ten inactive compounds, maintaining this balance even after dataset splitting. For example, with models that achieve an AUC of 0.85, we recommend using AUC as a metric only when the held-out test sets contain at least 20 active and inactive compounds[46]. When this is not feasible (for assays with few positive example compounds), AUC may not be an appropriate metric. Instead, we suggest evaluating models using absolute predictions instead of predicted probabilities, using metrics such as balanced accuracy, while carefully accounting for potential confounding effects from data imbalance using random prediction baselines and y-scrambling.

(5) Recognize the biological context of the assay before model interpretation. A general recommendation applicable beyond cytotoxicity is that insight into the relevant biological mechanisms and pathways can elucidate the reasons behind a model's performance. This involves recognizing typical cellular responses associated with the assay. We recommend interpreting the key features contributing to model performance[47,48]; for instance, mitochondrial granularity features are more predictive of mitochondrial toxicity than features from other imaging channels[49]. Additional factors influencing model interpretation include the choice of cell lines, treatment durations, and compound concentrations. Biological interpretability and simplicity should be weighted more heavily in computational model development.

## Benchmarking protein-target prediction with Cell Painting profiles compared cell counts

In order to facilitate the above recommendations, we are together with this work releasing a benchmark dataset for compound-activity

prediction curated along the recommendations above from EvEBio[50]. This benchmark dataset comprises ligand binding and functional assays in biochemical systems for nuclear receptors (NRs), alongside cell-based activity measurements for seven-transmembrane receptors (7TMs), supplemented by cell viability assays to monitor cytotoxicity, collectively covering a broad panel of these receptor classes. After applying the same filtering steps described above (see Benchmarking Cell Painting for curated Protein-Target data), we isolated a high-quality benchmark of 24 protein-target assays (including agonist and antagonist modes) with binary activity labels (active/inactive) across 269 compounds, which also have Cell Painting data publicly available in the Broad dataset. Using this benchmark, models leveraging the full Cell Painting profiles outperformed the baseline cell count model, achieving a higher mean balanced accuracy (BA = 0.67 vs. 0.52; see Fig. 7a). Compared to both a randomly shuffled baseline and the cell count baseline, the full-profile models delivered over 20% improvement in AUC-PR on 8 of the 24 tasks (Supplementary Fig. S4). Notably, gains in AUC-ROC were consistent regardless of the test set size or the proportion of active compounds, suggesting that these improvements are robust and not driven by dataset composition (Supplementary Fig. S4).

The assays most improved by including Cell Painting information were for GR agonism (mean AUC = 0.79 with the complete Cell Painting profile vs. AUC = 0.60 with cell count baseline model) and RARb antagonism (AUC = 0.83 vs. AUC = 0.64). The feature Nuclei_RadialDistribution_RadialCV_ER_4of4 was found to be the most predictive of this endpoint (Fig. 7b); this feature is the coefficient of variation of intensity in the ER channel within the outermost region of the nucleus object. For RARb antagonism, the most predictive feature was Nuclei_Texture_Entropy_Mito_10_0 (Fig. 7c), which measures the disorder or heterogeneity of the mitochondrial signal. Although mitochondria are cytoplasmic, limited image resolution and perinuclear overlap can cause mitochondrial signals to be captured within the nuclear mask. In both the GR agonist and RARβ antagonist cases, the phenotypes are subtle and not readily apparent by visual inspection (as confirmed in Supplementary Figs. S5 and S6, which show no substantial changes in cell count). However, these effects are best captured through image-derived features. The superior performance of models using the full Cell Painting profiles compared to cell count baselines further supports the idea that spatial and texture-based features carry the relevant biological signal.

Overall, the value of imaging-derived features largely hinges on the screening context. Previous studies have explored the importance of particular Cell Painting features with respect to mitochondrial toxicity (edge of the mitochondria object)[49], cardiotoxicity (fine-grained smoothness of the ER staining and size, shape, number, or texture of nucleoli within the nucleus)[51], and cytotoxicity prediction[52], particularly also in in muscle cells (spots edges and ridges profiles in the ER and mitochondria channels)[53]. Our results underscore the utility of the complete Cell Painting profile as high-content imaging data for capturing biologically meaningful phenotypes associated with signaling pathways, not just limited to cell count. These phenotypic responses become more detectable when cytotoxicity is not pervasive in the dataset, causing models to be focused on cell count over more subtle cellular changes.

## Limitations of this study and future directions

A key limitation of all previously existing benchmark datasets from publicly available bioactivity data is that they include assays that are off-target or secondary pharmacology screens rather than solely measurements against the compounds' intended primary targets. As a result, the compounds tested often share overlapping chemical features and target similar protein families, reducing the independence between assays and potentially inflating model performance due to correlated activity profiles. Further, the functional relevance of assay

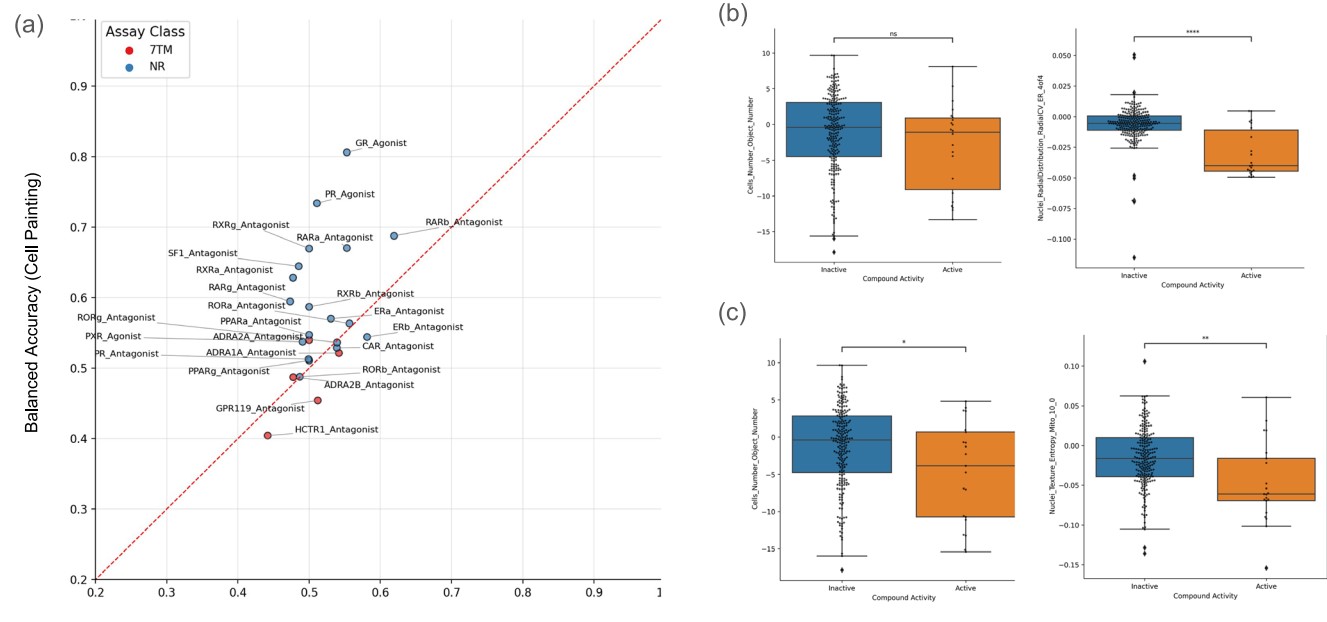

**Fig. 7 | Full Cell Painting profiles outperform cell count–only baselines across multiple biological assays and representative feature comparisons. a** Mean balanced accuracy for each of the 24 assays in the EveBio dataset comparing models trained on the full Cell Painting feature set versus a baseline model using only the cell-count feature. Data points represent independent biological replicates ($n = 24$ assays), where each assay constitutes a distinct biological unit evaluated independently. **b**, **c** Distribution of feature values for active and inactive compounds in two representative assays. **b** shows the cell count and Nuclei_RadialDistribution_RadialCV_ER_4of4 features for compounds classified in the GR_Agonist assay (active: $n = 22$; inactive: $n = 239$), and **c** shows the cell count and Nuclei_Texture_Entropy_Mito_10_0 features for compounds in the RARb_Antagonist assay (active: $n = 21$; inactive: $n = 224$). Each point represents an independent compound measurement, where each compound corresponds to a unique

chemical structure. Box plots show the median (center line), the first and third quartiles (bounds of the box), and whiskers extending to 1.5× the interquartile range; points outside this range are plotted as outliers. Statistical analysis. Statistical significance was assessed using a two-sided Mann–Whitney–Wilcoxon test with Bonferroni correction applied across assays. For the GR_Agonist assay, the cell-count yielded $U = 1.557e-01$ and $p = 3.110e+03$ (two-sided), and Nuclei_RadialDistribution_RadialCV_ER_4of4 features yielded $U = 4.252e+03$ and $p = 1.679e-06$ (two-sided). For the RARb_Antagonist assay, the cell-count feature yielded $U = 3.350e+03$ and $p = 1.317e-03$ (two-sided), while the Nuclei_Texture_Entropy_Mito_10_0 feature yielded $U = 3.046e+03$ and $p = 2.564e-02$ (two-sided). Significance thresholds are reported as ns: $5.00e-02 < p \le 1.00e+00$; *: $1.00e-02 < p \le 5.00e-02$; **: $1.00e-03 < p \le 1.00e-02$; ***: $1.00e-04 < p \le 1.00e-03$; ****: $p \le 1.00e-04$.

readouts can vary substantially depending on the compound's mode of action. For example, antagonists tested in the absence of an agonist may not elicit functional responses detectable by high-content imaging, while agonizts can induce receptor desensitization, influencing time-dependent phenotypes. These complexities limit the interpretability of morphological changes and challenge the ability to distinguish specific pharmacological mechanisms from indirect or off-target effects. Some compounds commonly annotated as GPCR actives may, in fact, be modulators of related pathways such as phosphodiesterases (e.g., IBMX), transporters, or metabolic enzymes[54]. Without well-controlled assay conditions, such compounds could be misclassified as receptor agonizts or antagonists, further confounding model training and evaluation. Despite efforts to curate the dataset (such as removing assays with excessive cytotoxicity and cell line inhibition), these confounding factors remain inherent to the current public datasets.

Across all datasets analyzed, the Broad, AstraZeneca, and New York Stem Cell Foundation Cell Painting datasets, we evaluated whether cell count alone could predict various assay outcomes or intended tasks set in the original studies (such as separating drugs from positive controls). In each dataset, we found that cell count performed comparably to models using full Cell Painting profiles. Our results consistently show that bioactive compounds, as defined in these benchmarks, tend to reduce cell count, especially when the assays in these benchmarks are biased toward viability-related endpoints.

We also acknowledge that technical artifacts, such as plate layout effects, can confound assay activity predictions, particularly when bioactive compounds cluster on specific plates. In the case of the Broad dataset, by analyzing plate-level data, we demonstrate

consistent reductions in cell count for bioactive compounds (as defined by current benchmarks) across plates, indicating that the predictive power of cell count is unlikely to be driven solely by technical bias. Furthermore, given that the Broad, AstraZeneca, and New York Stem Cell Foundation Cell Painting datasets' image sets were all independently created and not likely to follow the same plate and batch layouts, this rules out artifacts as yielding falsely positive results; for new datasets, the potential impact of plate and batch layout needs to be evaluated for each case separately.

Beyond cytotoxicity, additional sources of bias can influence predictive performance. These include cell line–specific responses, staining or imaging artifacts, and variability in experimental conditions or protocols (e.g., incubation times, compound dosing, or reagent preparation). While our cross-dataset analysis mitigates some of these issues, we recommend that studies explicitly evaluate such factors (e.g., following protocols like those of the OASIS[54]). In particular, chemicals should not be grouped by bioactivity; instead, randomized layouts are advisable. A key pitfall to avoid is running Cell Painting assays with the exact same plate layout as the target assay, since this allows technical artifacts to confound both readouts. Importantly, none of the datasets used in this study suffers from this issue.

Overall, in this study, we reveal a critical limitation in current assay prediction benchmarks: many are inadvertently confounded by viability assays, which can dominate predictive performance. By systematically evaluating cell count as a baseline, we show that it is often highly predictive of assay outcomes, particularly for viability-related endpoints, raising concerns about the benchmarks used to evaluate high-dimensional phenotypic profiles and complex machine learning

models. Our results suggest that some benchmarks may overstate model performance by favoring compounds with strong cytotoxic effects or dramatic morphological changes, which are easier to detect. This introduces a bias that undermines fair evaluation of models intended to capture more subtle phenotypic signals. To address this, future studies would benefit from the five actionable recommendations for the use of high-dimensional phenotyping readouts, including Cell Painting, transcriptomics, and proteomics, in machine learning workflows. These practices aim to promote more rigorous, interpretable, and biologically meaningful model evaluation. We also highlight promising future directions, such as excluding assays predictable by cell count from multi-task models or applying adversarial training to remove confounding signals. Our findings emphasize the need for more careful benchmarking to ensure progress in phenotypic profiling translates into real advances in predictive toxicology and drug discovery. Code and data are publicly available at https://github. com/srijitseal/The_Seal_Files and https://doi.org/10.5281/zenodo. 17168185, including a filtered Cell Painting dataset where features highly correlated to cell count were removed.

## Methods

### Cell Painting data
We used the Bray Cell Painting dataset[42] (cpg0012-wawer-bioactive-compoundprofiling) in this work, which contains images of cells treated with over 30,000 chemical perturbations. We used the dataset processed by Seal et al.[49], where for each plate, the average feature value of the DMSO controls was subtracted from the average feature value of the perturbations and then divided by the standard deviation for the DMSO samples on the plate. Finally, the median feature value was calculated for each compound and dose combination. For replicates, the median feature value was considered only for doses within one standard deviation of the mean dose across all perturbations of the same compound. Thus, the features from the Cell Painting dataset used in this study are the deviation from DMSO controls on the same plate after normalization, and after aggregating the median across replicates. We use the Cell Painting descriptor "Cells_Number_Object_Number", which correlates directly with cell count, as the sole feature in our cell-count baseline cell count models.

### Bioactivity data description
We next gathered three annotated assay benchmarking datasets from the literature: Moshkov et al.[13], Hofmarcher et al.[12], and Ha et al.[15] (Table 1). The Moshkov dataset contains bioactivity data from 270 assays, predominantly cell-based and toxicity-related (e.g., growth inhibition, viability) for 16,170 compounds, with 13.4% completeness and 2.7% active data points (defined from individual assay hitcalls). The Hofmarcher dataset, sourced from ChEMBL, comprises data from 209 assays, featuring quantitative pChEMBL data from individual protein targets, as well as tumor cell line growth inhibition assays, for 10,573 compounds, with 2.5% completeness and 34.7% active data points. The Ha dataset, an update on the Hofmarcher dataset, comprises data for 201 assays from ChEMBL, focusing on protein target activity for 10,526 compounds, with 2.6% completeness and 34.7% active data points. These datasets have been used before to evaluate novel machine learning algorithms predicting compound activity using Cell Painting descriptors, making them suitable for comparing our baseline cell count models[12–15].

### Comparing active with inactive compounds
Based on binary activity annotations in each of the three datasets, we compared the cell count distributions between active and inactive compounds for each assay using an independent t-test (as implemented in scipy.stats[55]). To mitigate bias from unequal group sizes, we sampled both groups to match the size of the smaller group. We then compared the distribution of the Cells_Number_Object_Number feature for active versus inactive compounds with a t-test for independent samples using the Bonferroni correction.

### Predictivity of cell count
We systematically assessed the predictive utility of the cell count feature across multiple assays by using the raw cell count as a prediction score to understand its predictive power by evaluating classification performance at different thresholds. For each assay across each of the three datasets, binary predictions were generated using a probabilistic scoring approach (logistic regression with fixed weight of −1) based on the Cells_Number_Object_Number feature. A sigmoid-like function (Eq. 1) was applied to transform feature values into probability scores, mapping the difference between each value and a specified threshold to a probability score between 0 and 1.

$$Probability\ Score = \frac{1}{1 + e^{(v - t)}} \qquad (1)$$

iWhere $v$ is the raw feature value for a specific compound, in this case, the Cells_Number_Object_Number, and $t$ is the chosen threshold used to center the transformation. Equation 1 thus assigns higher probabilities to compounds with lower cell counts, consistent with cytotoxic effects.

We then evaluated how varying thresholds on the cell count distributions impacted the ability to predict assay outcomes. The separation of classes was assessed using balanced accuracy at each threshold (probability scores greater than or equal to 0.5 as positive and those below as negative), emphasizing the ability of the cell count feature to distinguish positive from negative classes. Since our goal was solely to separate classes and not to train and test an ML model, a low balanced accuracy score of 0.10 was interpreted as 0.90, given that the thresholding model is fixed to classify compounds that decrease cell count as active (and often makes opposite predictions). This approach emphasizes our focus on class separation rather than absolute predictions, meaning the assay-specific definition of activity (e.g., whether increased cell counts indicate activity) does not affect our interpretation.

### Predicting compound activity from phenotypic profiles
The Moshkov dataset[13] uses binary classification across 270 tasks, with scaffold-based splits for training and testing defined by the original authors. We excluded 40 assays, which did not contain enough data points to guarantee that at least one active and one inactive compound is included in the training and test data that Moshkov et al. used for each cross-validation fold. A logistic regression classifier, using sklearn's default hyperparameters, was trained using "Cells_Number_Object_Number" as the sole feature to predict each binary endpoint. For comparison, a separate logistic regression model was trained using all CellProfiler features for each endpoint. L2 regularization with $C = 0.001$ and balanced class weighting parameters ensured the model handled the high feature dimensionality and class imbalance well. Five-fold cross-validation was performed using the splits provided by the authors, and performance was measured using the Area Under the Receiver Operating Characteristic Curve (AUC) in combination with the Area Under the Precision-Recall-Gain Curve (AUPRG)[56]. AUPRG was selected as an additional metric due to the large class imbalances in most of the endpoints within the Moshkov dataset, where 232 of the original 270 tasks had at least five times more inactive than active compounds, and 185 had at least a 10:1 inactive-to-active ratio. AUPRG provides a universal baseline, which is comparable across assays with different class distributions (unlike Area Under the Precision-Recall Curve, AUCPR), and is less impacted than AUC by the abundance of negative class examples that are present in these classification tasks. For other tasks, we limited the analysis to AUC-ROC to enable direct comparison to prior published results.

## Comparing the prediction of biological assays between Cell Painting and convolutional networks

We used the Hofmarcher dataset with the same train-validation-test datasets as in the original publication (70-10-20 splits repeated three times for 209 tasks)[12]. An XGBoost classifier (chosen for the sake of comparability with the original paper) was trained for each assay using only the cell count feature, with hyperparameter optimization performed via randomized search across a predefined grid in a 5-fold stratified cross-validation. The baseline cell count models were trained on the 80% training set and evaluated on the 20% test set. This process was repeated for all three train-test splits provided by the authors. Model performance was assessed using the AUC, allowing for direct comparison with results from Hofmarcher et al.

## Comparison to contrastive learning (CLOOME): bioactivity prediction

We next compared the performance of a logistic regression (LR) model trained on the cell count feature to Sanchez-Fernandez et al.[14], who trained an LR model on CLOOME embeddings for each of the 209 targets in the Hofmarcher dataset to compare feature spaces like-to-like, while keeping all other parameters identical. Since CLOOME embeddings have access to chemical structures, two baseline cell count models were considered: (1) using cell count as the sole feature and (2) using cell count along with molecular weight and logP (calculated using RDKit) as very simple chemical features. The first baseline excludes any chemical information, while the second baseline, although not explicitly trained on chemical features, incorporates some chemical-related information indirectly through molecular weight and logP. Molecular weight and logP were chosen as proxies for chemical information, as these are known confounding factors in the chemical assays[57,58]. Employing the same 70-10-20 data splits, multiple logistic regression models were trained per assay with varying regularization strengths, selecting the best based on validation AUC. Optimal models were then evaluated on the test set. This process was repeated for all three train-test splits provided by the authors, with evaluations based on the mean AUC for each assay.

## Comparing enrichment in gene ontology for selected compounds

We performed two Gene Ontology (GO) enrichment analyses to characterize the biological processes associated with compound-induced transcriptional changes. This work was built on a previous study[59], in which a framework was established to connect transcriptional and morphological perturbations.

Transcriptomic signatures were obtained from the LINCS L1000 dataset (level5_beta_trt_cp_n720216x12328.gctx) using the signatureSearch R package (v1.12.0)[60]. First, we analyzed compounds for which Cells_Number_Object_Number values are relatively low compared to controls in the Cell Painting assays. Second, we focused on promiscuous compounds, where we combined the Moshkov and Hofmarcher datasets and stratified compounds by the number of assays in which they were labeled active. Among the 1520 compounds in Cell Painting that have a value of Cells_Number_Object_Number lower than −14.35 (min = −50.59), 534 were also present in the LINCS resource and passed the is_exemplar filter (which identifies a single "exemplar" signature for each perturbagen in each cell line). Of these, 466 compounds were tested in U2OS ($n = 383$), MCF7 ($n = 73$), or PC3 ($n = 10$) cell lines. Cell line selection followed the priority order U2OS > MCF7 > PC3, based on our prior demonstration that these cell lines share over 80% of genes with U2OS[59]. For the analysis on promiscuous compounds, starting with 24,606 labeled compounds (0–86 active labels per compound across the combined Hofmarcher and Moshkov datasets), we identified 8082 compounds present in LINCS that passed the is_exemplar filter. Of these, 4094 were tested in U2OS ($n = 3,630$), MCF7 ($n = 434$), or PC3 ($n = 33$) cell lines. We restricted the analysis to

the landmark gene set, which represents directly measured genes (978 genes) in the L1000 assay and provides the most reliable information content. For each compound, perturbed genes were selected using a z-score threshold ($|z| > 2$). GO enrichment was performed with the clusterProfiler R package (v4.6.2), using the enrichGO function and the org.Hs.eg.db annotation database. Parameters were set as follows: pAdjustMethod = "bonferroni" and pvalueCutoff = 0.05.

## Few-shot learning for molecule activity prediction

Subsequently, we compared baseline cell count models to those developed by Ha et al., who used 161 assays for training a few-shot learning model and 18 separate assays as the test set[15]. Since no training data were available for the test assays, the baseline cell count model was trained by combining the outcomes of all 161 training assays into a single endpoint. This approach encourages the model to capture general trends rather than assay-specific information, aligning with our hypothesis that this generic information might primarily reflect cell death. Hyperparameter optimization for the XGBoost model was performed using a 5-fold stratified cross-validation strategy, with cell count as the sole feature. Model performance was evaluated individually on each of the 18 test tasks designated by the authors, using AUC as the performance metric.

## Comparison to label-free Cell Painting profiles from brightfield images

We next used the test dataset provided in Cross-Zamirski et al.[41], which included 77 negative DMSO wells, 26 positive wells, and 170 compound perturbation wells. We used 4 feature spaces, namely (a) Cell Painting profiles from the ground truth, (b) features from predicted images from the conditional Wasserstein Generative Adversarial Network with gradient penalty (cWGAN-GP) model, (c) features from predicted images from the U-Net model as released by the authors, and (d) the baseline cell count model using the Cells_Neighbors_SecondClosestObjectNumber_5 feature from the ground truth profiles. To identify compounds with phenotypic similarity to positive controls, we first used a k-Nearest Neighbors (k-NN) classifier ($k = 5$) with Euclidean distance in each of the above feature spaces to distinguish between the positive and negative controls. The model was trained and evaluated through 100 iterations, where the training data was randomly resampled in each iteration to ensure balanced representation of both classes. Predictions for each of the 170 compound perturbation wells were aggregated using majority voting across all iterations, identifying those test compounds predicted as the positive control.

## Comparison to feature extraction from high-content imaging screens

We next used the dataset released by Comolet et al.[44], which consisted of fibroblasts from 20 donors treated with three compounds (CP21R7, Pemigatinib, and Y-39983-HCl) at two concentrations (0.2 and 1.0 μM) each in order to compare the ability of various Cell Painting feature space to identify compounds from controls. The authors performed confounder-correction (plates, wells, rows, columns, donor), normalized between 0 and 1, and averaged at the well level to mitigate noise and emphasize the predominant effects observed across the majority of the cell population[44]. We used two feature spaces to compare the ability to separate perturbations from DMSO controls—first, the ScaleFEx's features, which included all extracted morphological and phenotypic descriptors from cell imaging assays and second, only the MitoCount feature extracted by ScaleFEx, which was the closest related feature to cell count[44]. We reproduced the methods described by Comolet et al. to assess the accuracy of a logistic regression (LR) model designed to distinguish each drug from the control (DMSO). For each drug and concentration, a balanced training set was created by sampling DMSO controls to match the minority class. For models using ScaleFEx features, a Recursive Feature Elimination (RFE) and logistic

regression model was employed to identify the most predictive, non-redundant features for classification. For the baseline cell count model using MitoCount, no feature selection was used. A logistic regression model was trained and validated using leave-one-plate-out cross-validation (which results in information leak of compounds which was spread across all plates), that is, in each fold, training is performed on all but one plate, which is held out for testing. The models are trained iteratively to classify wells treated with each drug and DMSO controls. The values were corrected for plate, donor, column, and row effects and normalized to a range of 0–1. A Mann–Whitney U test was used to assess statistical differences between a drug and DMSO control, with Bonferroni adjustment applied to account for multiple comparisons.

### Evaluating the effects of concentration-dose response in Cell Painting

We hypothesized that the signal in Cell Painting depends on compound concentration and that single concentration data available in public datasets may not always be optimal in detecting the morphological changes that ML models can learn from. As an illustrative example, we generated dose-response Cell Painting data for six compounds: aminohexylgeldanamycin (Hsp90 inhibitor), givinostat (HDAC inhibitor), repotrectinib (tyrosine kinase inhibitor), I-BET282 (pan-inhibitor of all eight BET bromodomains), bisindolylmaleimide V (protein kinase C inhibitor), and PP58 (Src inhibitor), in 2 replicates and ten doses.

### Benchmarking Cell Painting for curated protein-target data

To evaluate the use of Cell Painting assay to predict protein targets of compounds, we curated a benchmark dataset using the recently released data from EvE Bio (Release #4 as of May 17, 2025)[61], covering both agonist and antagonist modes across nuclear receptors and 7-transmembrane receptors. This dataset brings the total screened combinations to approximately 237,490 compound–target pairs, following a two-phase quantitative protocol: an initial screen at three high concentrations: (0.6, 2.5 and 10 μM), in duplicate, followed by a higher-resolution profiling phase for compounds meeting progression criteria. Activity assignments are determined algorithmically based on curve fitting, potency metrics (pXC50), and gating rules across receptor subfamilies; compounds flagged for promiscuity are annotated. We mapped this EvE dataset to available Cell Painting profiles, applied similar quality controls as above (removing promiscuous compounds and compounds that exhibit high viability, and likely inactive compounds described in the original work), and excluded viability assays. For Cell Painting data, compounds with low cell count, defined as mean minus two standard deviations, were filtered out, and assays lacking sufficient class diversity (<20 compounds per class) were removed. These filtering steps yielded a high-quality benchmark of 24 protein-target assays (including agonist and antagonist modes) with binary activity labels (active/inactive) across 269 compounds.

We evaluated the predictive performance of two feature sets to predict protein targets: the full Cell Painting profiles extracted using CellProfiler and a baseline cell count model. The baseline model used a single morphological feature representing cell count, while the Cell Painting feature set encompassed diverse cellular morphology descriptors. To address class imbalance, class weights were calculated using the balanced strategy (classes with fewer samples receive higher weights), ensuring equal contribution of positive and negative classes during model training. Models were trained using stratified 3-fold nested cross-validation to maintain balanced class distributions in both training and test sets. Logistic regression was employed for the baseline cell count model with the cell count feature, while Random Forest classifiers were used for the multi-dimensional Cell Painting features. For hyperparameter optimization, 2/3rds of the data was used in the inner cross-validation loop, applying RandomizedSearchCV to tune parameters. This included regularization

strength for logistic regression and tree depth, leaf size, and the number of estimators for Random Forests. The best-performing models were selected based on AUC-ROC scores from the inner cross-validation loop. Classification thresholds were optimized using Cohen's Kappa, evaluated across a range of probability thresholds to maximize agreement between predicted and true labels in the training data. Final model performance was assessed on the outer test set using metrics such as Balanced Accuracy, AUC-ROC, AUC-PR, Matthews Correlation Coefficient (MCC), and Cohen's Kappa. To ensure robustness, random label scrambling was performed for each assay, and models were retrained on the scrambled labels to confirm that their performance was not driven by chance[62].

### Statistics and reproducibility

All statistical analyses were performed as implemented in scikit-learn[63]. All code is available at https://github.com/srijitseal/The_Seal_Files. All data and code are released via https://doi.org/10.5281/zenodo.17168185[64,65].

### Reporting summary

Further information on research design is available in the Nature Portfolio Reporting Summary linked to this article.

## Data availability

The data used in this study have been deposited in the Zenodo database under accession code https://doi.org/10.5281/zenodo.17168185 [https://doi.org/10.5281/zenodo.14838603]. Supplementary Data 1 provides annotated assays from the Hofmarcher dataset, along with associated metadata, including assay type, target type, organism, and assay category. Supplementary Data 2 provides a comparison of the Protonet CP+ model (at a support set size of 64) with the baseline cell count model, as benchmarked on FSL-CP in Ha et al. Figure 2e-h can be reproduced with source data from Supplementary Data 3. All other Figures can be reproduced from data and notebooks deposited at Zenodo. Supplementary data are provided with this paper.

## Code availability

The code used to develop the model, perform the analyses, and generate results in this study is publicly available and has been deposited in Zenodo at https://doi.org/10.5281/zenodo.1793112465 (and GitHub at https://github.com/srijitseal/The_Seal_Files), under MIT license. This includes a notebook with steps to filter the Cell Painting dataset for features highly correlated to cell count: https://github.com/srijitseal/The_Seal_Files/blob/main/02_Remove_Confounders_Cell_Painting_Dataset.ipynb. The specific version of the code associated with this publication is archived in Zenodo and is accessible via https://doi.org/10.5281/zenodo.14838603[64].

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

## Acknowledgements

S. Seal acknowledges Reid Olsen (Recursion), Anna Lobley (Independent), Barak Gilboa (Novo Nordisk), and Hassan A Ali (University of Miami) for their comments on a pre-print and to the independent reviewers for their suggested analyses that greatly enhanced this publication. S. Seal acknowledges funding from the Cambridge Centre for Data-Driven Discovery (C2D3) Accelerate Programme for Scientific Discovery. S. Seal, S. Singh, and A.E.C. acknowledge funding from the National Institutes of Health (NIH MIRA R35 GM122547 to A.E.C.), the Massachusetts Life Sciences Center Bits to Bytes Capital Call program for funding the data production (to S. Singh), as well as their Data Science Internship program (to S. Singh), and the OASIS Consortium organized by HESI. O.S. acknowledges funding from the Swedish Research Council (grants 2020-03731, 2020-01865, 2024-03566, 2024-04576), FORMAS (grant 2022-00940), Swedish Cancer Foundation (22 2412 Pj 03 H), and Horizon Europe grant agreement #101057014 (PARC) and #101057442 (REMEDI4ALL). W. Dee and G. Slabaugh acknowledge the UKRI/BBSRC Collaborative Training Partnership in AI for Drug Discovery, led by Exscientia Plc. in partnership with Queen Mary University of London. The Collaborative Training Partnership was funded by the Biotechnology and Biological Sciences Research Council, grant reference BB/X511791/1. G. Slabaugh also acknowledges EPSRC grant EP/Y009800/1, through Keystone project funding from Responsible AI UK (KP0016) and also acknowledges the support of the National Institute for Health and Care Research Barts Biomedical Research Centre (NIHR203330); a delivery partnership of Barts Health NHS Trust, Queen Mary University of London, St George's University Hospitals NHS Foundation Trust and St George's University of London.

## Author contributions

S. Seal conceived the study, designed all analyses, performed the modeling, carried out benchmarking and data interpretation, curated datasets, generated figures, and wrote the manuscript. W.D. ran several model comparisons and co-wrote the manuscript with S. Seal. A.S. supported code development and preprocessing of Cell Painting data. N.C. contributed to the gene expression analyses. A.Z. helped with code and data handling. E.M. assisted in the analysis of selected Cell Painting images. K.T., Á.A.C., D.B., and A. Beatson contributed on behalf of Axiom Bio to describe the experiments involving cell count versus concentration data. G.S. provided comments on the manuscript and guidance on model evaluation. O.T. and J.C.P. assisted in writing the manuscript. S. Singh provided expertise on Cell Painting data processing and interpretation. O.S., A. Bender, and A.E.C. co-supervised the work together with S. Seal, advising throughout study design, model interpretation, and manuscript writing and revision. All authors reviewed, contributed to, and approved the final manuscript.

## Funding

## Competing interests

S. Singh and A.E.C. serve as scientific advisors for companies that use image-based profiling and Cell Painting (A.E.C.: Recursion, SyzOnc, Quiver Bioscience; S. Singh: Waypoint Bio, Dewpoint Therapeutics, DeepCell) and receive honoraria for occasional talks at pharmaceutical and biotechnology companies. J.C.P. and O.S. declare ownership in Phenaros Pharmaceuticals. G.S. serves as a scientific advisor to BioAI-Health and has a collaborative project with AstraZeneca involving image-based profiling and Cell Painting. The remaining authors declare no competing interests.
