## [Transparent Peer Review file · Nature Communications]

Counting Cells Can Accurately Predict Small-Molecule Bioactivity Benchmarks

Corresponding Author: Dr Srijit Seal

Version 0:

Reviewer comments:

Reviewer #1

(Remarks to the Author)

The authors investigate the role of cell count as a simple baseline feature in predicting small molecule bioactivity within the context of phenotypic profiling methods, such as Cell Painting and gene expression profiling. The authors argue that many widely used benchmark datasets in drug discovery are biased toward cell viability or cytotoxicity assays, where a basic metric like cell count can perform comparably to more complex, high-dimensional phenotypic profiles. By analyzing multiple datasets and introducing a new curated benchmark, the study proposes using cell count as a baseline to assess the added value of sophisticated profiling techniques. It provides actionable recommendations for improving machine learning model design and evaluation in this domain.

The paper is well-written and very extensive in its assessments. Overall, this work contributes to a more rigorous understanding of how machine learning models should be evaluated in drug discovery in the context of Cell Painting and gene expression.

However, I have some concerns regarding the study:

- While the abstract mentions gene expression alongside Cell Painting, the analysis heavily emphasizes image-based profiling. A more detailed comparison with gene expression data (e.g., in the Moshkov dataset) could strengthen the generalizability of the findings across phenotypic profiling methods.
- It seems to me that the study primarily focuses on datasets that are known to contain a high proportion of cytotoxicity-related assays. It would be beneficial to extend the analysis to datasets that include more diverse bioactivity endpoints. Does their conclusion also apply to other datasets with less cytotoxic compounds?
- Please discuss other potential biases beyond cytotoxicity (e.g., cell line effects, staining artifacts, variations in experimental conditions or protocols, etc.).
- While the paper highlights the predictive power of cell count, it does not explore other potentially useful simple features, such as nuclear morphology or mitochondrial integrity. Are these also correlated with the endpoint?
- I would be important if the authors provided a way to filter out datasets to remove associations highly correlated to cell count. Also, please subset existing datasets to remove this potential bias and provide this as a contribution to the community.
- The paper argues that cell count explains much of the predictive power, but it does not fully dissect whether specific Cell Painting features provide additional mechanistic insights beyond viability.
- The manuscript's analysis of concentration-dependent effects is compelling but restricted to only six compounds. This small sample size undermines the generalizability of the findings and the claim that multi-dose experiments are critical for robust phenotypic profiling.
- There is an unclear breakdown of assay types. The authors note that many benchmark essays are related to cell viability or cytotoxicity, but they do not quantify or categorize the assay types across datasets. Could they provide a table that breaks down the assay type in each dataset? (e.g., viability, protein-target, etc.)
- The manuscript employs different baseline models (e.g., logistic regression, XGBoost) across analyses, which may be confusing about the primary baseline being advocated and complicate comparisons. XGBoost with a single feature is an overkill. Can the authors also use the raw cell count as a prediction score to understand its prediction power?
- Why did the authors use only molecular weight and logP as chemical features? What is the rationale behind choosing only these?

(Remarks on code availability)

Reviewer #2

(Remarks to the Author)

This is an interesting and valuable study that challenges prevailing assumptions in phenotypic screening by showing that simple cell count features can perform comparably to more complex morphological or transcriptomic features (e.g., Cell Painting or gene expression) in predicting compound bioactivity. The manuscript is well-written and addresses a timely and relevant question in the field of machine learning for drug discovery. I appreciate the authors' efforts in systematically reevaluating benchmark datasets and proposing practical recommendations. However, I believe the manuscript would benefit from addressing the following points before publication:

1. On the specificity of cell count as a proxy for toxicity:

The authors argue that many assays are primarily capturing cell health or cytotoxicity signals, which may explain why cell count alone performs well. However, cytotoxicity is not binary—it can vary in intensity and specificity. Could cell count only detect strong cytotoxic effects while missing more nuanced or phenotype-specific signals? I encourage the authors to further clarify the scope and limits of what cell count can capture, perhaps by stratifying compounds by degree or type of toxicity and evaluating the cell count model's sensitivity.

2. On the complementarity between deep learning models and cell count models:

While the authors compare cell count with several state-of-the-art models, the current manuscript does not analyze what differences exist between the compounds identified as active by different methods. A Venn diagram is not enough. I suggest the authors perform a deeper analysis of the compounds that are uniquely identified by deep learning models but missed by cell count models. What are the chemical or functional characteristics of these compounds? It is possible that these uniquely identified compounds have more subtle or complex modes of action, and understanding this may significantly strengthen the paper's conclusions.

3. On clarity of the conclusions:

Although the authors have conducted extensive experiments, the conclusions—particularly in the discussion section—are not sufficiently distilled. I suggest the authors revise the discussion to clearly articulate their key findings, implications for future benchmarking practices, and the broader impact of their work on method development in the field.

4. On model simplicity and interpretability:

The authors make an important point regarding the value of simple and interpretable methods in bioactivity prediction. This aligns with recent findings in other domains of computational biology. For example, SpaDo (Genome Biology, 2024) showed that in spatial transcriptomics, simple clustering based on spatial neighbor cell type proportions could achieve comparable or better results than complex graph neural network models. I recommend that the authors cite such work to further reinforce the message that biological interpretability and simplicity should be weighted more heavily in computational model development.

(Remarks on code availability)

Reviewer #3

(Remarks to the Author)

Summary

This manuscript presents an analysis of assay predictability from image-based phenotypic assays, specifically the Cell Painting assay, and the role of cell count as a feature that can drive performance. The problem of assay prediction is very complex due to the multiple data sources that interact and the many confounding factors that can lead to incorrect interpretations. The analysis and effort presented by the authors is a valuable step forward to improve clarity about how this problem can be approached and what corrections need to be done going forward to advance the state of the art in this field.

While the questions and presented analysis are promising, the main limitation of this study is that it continues to rely primarily on the same phenotypic dataset, which may no longer be useful to make progress in assay prediction problems. The study makes the assumption that cell count is a clean readout, while in practice, the dataset used may have significant technical artifacts due to plate layouts and batch effects. The pattern reported in this paper may be a more fundamental issue with the imaging dataset instead of being a generalized phenotypic property for assay prediction. In a related note, the assay endpoints across the three benchmark sets analyzed may all be related to each other because they may activate the same (polypharmacological) compounds. These problems are part of the challenges of the assay prediction problem, and need to be addressed in this study to more convincingly support the conclusions.

Main comments

1. Figure 1 shows cell count distributions and Figure 2 shows how many assays can be predicted with cell counts relative to other predictors. The main question is: are cell counts batch-effect independent? Or are cell counts biased by batch effects as much as other features? It is well known that cell density is a significant source of technical (not phenotypic) variation, and the cpg0012-wawer-bioactivecompoundprofiling dataset may have significant batch biases, because bioactive

compounds were placed in certain plates different from those used for non bioactive compounds. How are cell counts correlated with plates or batches? A potential answer could be obtained by looking at the distribution of cell counts in control well only, but other types of analysis may be needed to clarify this question.

2. If artifacts that affect cell count exist, how can we make sure that the predictive power of cell counts is not an artifact, but rather a real phenotypic effect? The main concern is that all the predictors (including cell counts) may be leveraging technical variation to predict the same subset of bioactive compounds (Fig 2.), which may come from the same plates (i.e., same technical artifact).

3. Fig. 1 shows similar results in three datasets, but in reality, it's the same phenotypic dataset. In other words, the cell counts come from the same imaging dataset. The difference between the three datasets is the assays only. In fact, the Hofmarcher and Ha datasets appear to be almost identical except for a handful of assays, according to the text and references. Is there any relationship between the Moshkov and Hofmarcher assays? Are the compounds labeled as active in Moshkov also active in Hofmarcher? Ultimately, it would be best to unify the set of assays into a single coherent benchmark. The presentation of results in three datasets (Moshkov, Hofmarcher, and Ha) is artificial, and makes it look as if the problem is widespread, but ultimately we are talking about the same phenotypic dataset. A unified view should be presented, with similar technical choices, such as cross validation differences and so on.

4. To confirm the claim more broadly, another phenotypic dataset should be analyzed as well. The manuscript mentions the JUMP dataset, for instance, which could be used as an independent source of phenotypic variation to investigate the role of cell counts for assay prediction. If the same compounds are present across the dataset, these can help to further investigate if the patterns observed in Fig. 1 and others are prevalent beyond the dataset that the community has used. Plate layouts will be different, and technical variation may have different biases which can lead to different results. The paper presents preliminary results with another set in Figure 8, but its scope is limited to support the broader claims.

5. Continuing with the batch effects issue in the selected dataset, are the models labeled as "All Features – Logistic Regression" in Figures 5 and 6 batch corrected? While the architecture is simpler, the features may be contaminated by technical variation that reveals where the samples come from (bioactive plates) rather than finding real morphological signals.

6. In a separate note, one of the recommendations for benchmarking is to keep assays that have sufficient positive and negative examples (more than 40). However, in many assay prediction problems, collecting such a high number of positive examples is unrealistic or simply defeats the purpose of predictive modeling because 40 hits may already be sufficient to move a drug discovery project forward. While having a high number of positive and negative examples is desirable for machine learning researchers, how can the authors include realistic, rare hit assays in their benchmarks? It is highly recommended to consider such examples, which may be more interesting for the community in practice.

Other comments:

7. Figure 1(a-c): are the points compounds or assays? If compounds, it seems too few points with respect to what's reported in the text (10K to 16K). If assays, it seems too many points with respect to what the caption says (~200). How were these points selected for the plot?

8. Are the phenotypes presented in Figure 4 consistent across replicates?

9. Lines 203 to 212 assert that AUPRG has benefits over AUPRC and AUC, giving the impression that the metrics were going to be unified and reported with the suggested metric. However, the results in the main manuscript are all based on AUC-ROC only. Why is the AUC-ROC adopted in this manuscript if another metric may have better interpretation properties?

(Remarks on code availability)

Version 1:

Reviewer comments:

Reviewer #1

(Remarks to the Author)

Thank you for carefully addressing all the comments and suggestions from my previous review. I have gone through the revised version of your manuscript, and I am pleased to see that my concerns have been resolved. The revisions have strengthened the paper and improved its overall clarity and impact.

I have no further comments and recommend the manuscript for publication.

(Remarks on code availability)

I checked the repository but did not run the code.

Reviewer #2

(Remarks to the Author)

The manuscript has been substantially improved, and most previous concerns have been fully addressed. The additional analyses greatly enhance the clarity and rigor of the work.

One minor issue remains: while Figure 1g clarifies that cell count captures strong and promiscuous cytotoxic responses, the manuscript still does not explicitly discuss its limited sensitivity to subtle or mechanism-specific effects. I suggest adding a brief statement in the Discussion to acknowledge this limitation.

With this minor revision, the paper will be suitable for publication.

(Remarks on code availability)

Reviewer #4

(Remarks to the Author)

Authors have performed a comprehensive and timely benchmark study about cell counting and Cell Painting verified from multiple aspects. The manuscript is well-organized and the results are comprehensive and convinced. The editor contacted me and asked me to review the author's responses to Reviewer #3's concerns. Authors have completely addressed the concerns proposed by Reviewer #3. The revised manuscript is well-written and the response letter is detailed. I have no additional suggestions.

(Remarks on code availability)

Response to review:

Small molecule bioactivity benchmarks are often well-predicted by counting cells

We sincerely thank the reviewers and editor for their thoughtful and detailed feedback. We are delighted that the reviewers found our manuscript well-written, timely, and a valuable contribution to the field. We added analyses and have revised the manuscript extensively in response to the comments and summarize our responses and changes below. All changes are marked in the revised manuscript and supplemental files.

REVIEWER COMMENTS

Reviewer #1 (Remarks to the Author):

The authors investigate the role of cell count as a simple baseline feature in predicting small-molecule bioactivity within the context of phenotypic profiling methods, such as Cell Painting and gene expression profiling. The authors argue that many widely used benchmark datasets in drug discovery are biased toward cell viability or cytotoxicity assays, where a basic metric like cell count can perform comparably to more complex, high-dimensional phenotypic profiles. By analyzing multiple datasets and introducing a new curated benchmark, the study proposes using cell count as a baseline to assess the added value of sophisticated profiling techniques. It provides actionable recommendations for improving machine learning model design and evaluation in this domain.

The paper is well-written and very extensive in its assessments. Overall, this work contributes to a more rigorous understanding of how machine learning models should be evaluated in drug discovery in the context of Cell Painting and gene expression.

However, I have some concerns regarding the study:

- While the abstract mentions gene expression alongside Cell Painting, the analysis heavily emphasizes image-based profiling. A more detailed comparison with gene expression data (e.g., in the Moshkov dataset) could strengthen the generalizability of the findings across phenotypic profiling methods.

We have now added a new analysis titled “Gene Enrichment for Cytotoxic and Promiscuous Compounds”:

We hypothesized that (1) a decrease in cell number (Cells_Number_Object_Number) observed in Cell Painting assays may reflect increased apoptotic activity, and (2) compound promiscuity may arise more from general cytotoxicity (e.g., apoptosis) than from selective activity. To investigate this, we leveraged gene expression data from the LINCS L1000 resource to examine the biological processes associated with compound perturbations. For this analysis, we selected only those compounds from LINCS that passed the “is_exemplar”

(which identifies a single ‘exemplar’ signature for each perturbation in each cell line) quality filter and were tested in U2OS, MCF7, or PC3 cell lines. Of the 1,520 compounds that significantly reduced cell count in the Broad Cell Painting assay, 466 had corresponding LINCS mRNA signatures, yielding a total of 550 gene expression profiles.

Gene Ontology (GO) enrichment analysis of genes with a z-score threshold ($|z| > 2$) revealed that transcriptional changes that were induced by compounds known to reduce cell count (by Cell Painting) were consistently associated with particular biologically relevant processes. The most frequently enriched pathways included the intrinsic apoptotic signaling pathway ($n = 112$ compounds), mitotic cell cycle phase transitions ($n = 101$), regulation of cyclin-dependent kinase activity, spindle organization, and nuclear division. Stress response pathways (e.g., reactive oxygen species, hypoxia) and metabolic functions (e.g., cholesterol/sterol biosynthesis) were also enriched.

Analyzing more carefully the degree of cell count impact, we found that compounds more strongly reducing cell count in Cell Painting assays showed the highest enrichment for apoptosis- and cell death-related GO terms (Figure 4a). Over 70% of compounds in the lowest cell count bins ($[-37; -50]$, normalized feature values against plate-specific DMSO controls) were linked to apoptosis-related terms, with this proportion decreasing as cell count increased. This underscores the importance of cell health and cytotoxicity as major drivers of both morphological and transcriptomic profiles, and highlights the need to account for cell count when benchmarking both phenotypic profiling methods.

We next explored relationships between gene expression and compound promiscuity (activity across many assays). From the set of 24,606 compounds with target activity labels (from Hofmarcher and Moskhov datasets), we extracted 8,381 LINCS signatures for 4,094 compounds that passed the cell line and exemplar filters. Again, the intrinsic apoptotic signaling pathway was the most frequently enriched category ($n = 193$), supporting our hypothesis that compound promiscuity is often associated with general cytotoxic effects (Figure 4b).

Figure 4. (a) Proportion of compounds enriched for apoptosis/cell death (red), other GO terms (blue), or no significant GO terms (gray), stratified by normalized cell counts bins, standardised against plate-specific DMSO controls. Compound counts per bin are shown above bars. Apoptosis-related enrichment exceeded 70% in two of the three lowest bins and declined with increasing cell count. (b) Similar analysis of compound promiscuity (i.e., number of active target labels for each compound). Apoptosis enrichment generally increases among compounds with an increase of active labels. (c) Euler diagram showing overlap between cytotoxic (n=1,520; orange) and promiscuous (n=24,606; blue) compounds, before (light) and after (dark) GO enrichment filtering. 214 compounds were shared between the two enriched subsets. (d) Heatmap showing apoptosis enrichment (%) across combinations of cell count bins and promiscuity bins. Each tile indicates the percentage and count (apoptosis / total). Color intensity reflects the level of apoptotic enrichment.

Beyond apoptosis, other recurrently enriched pathways associated with compound promiscuity included metabolic and biosynthetic processes (e.g., cholesterol/sterol biosynthesis [n = 164–156], secondary alcohol metabolism [n = 164–124], steroid biosynthesis [n = 112]), protein phosphorylation and kinase regulation (e.g., serine/threonine kinase activity [n = 161], protein kinase complex [n = 131]), and cell adhesion-related

categories (e.g., cadherin binding [n = 116], focal adhesion [n = 89]). Stress-related responses to ROS (n = 101), xenobiotics (n = 86), and hypoxia (n = 84) were also commonly enriched, highlighting the broad cellular impact of promiscuous compounds.

Focusing on the 214 compounds that reduced cell count and were also promiscuous (Figure 4c), we found that apoptotic enrichment increased markedly as cell count decreased. In contrast, the effect of promiscuity on apoptosis enrichment was more modest (Figure 4d). Notably, compounds with more than one active hit and a normalized cell count below -40 showed over 70% enrichment for apoptosis-related GO terms.

We have updated our Abstract to include these findings:

“We found that compounds that reduce cell count and exhibit promiscuous activity often trigger apoptosis-related gene expression, suggesting general cytotoxicity, rather than selective targeting, drives their biological impact.”

Our fundamental finding is that cell count is a simple baseline that can predict assay outcomes as well as Cell Painting can, given the current flawed benchmarks which are discussed in the paper. Other studies have already compared Cell Painting and gene expression in assay prediction [9, 18, 19, 20] so we now discuss these in the Introduction section (see text below). We also emphasize our finding in this study that 63% (31/49) of the assays in the Moshkov datasets which were well-predicted by gene expression were also well-predicted by cell count (Figure 2c), which can be largely attributed to the shared cytotoxicity signal. Finally, we add a section to the introduction describing the more general issue in machine learning for biology, that complex models are often beat by simple baselines:

“Previous studies have benchmarked models that make other kinds of predictions using gene expression data.[18] For example, Csentesi et al. found that simple baselines outperformed state-of-the-art foundation models in predicting post-perturbation gene expression.[19] Similarly, SpaDo showed that in spatial transcriptomics, simple clustering based on spatial neighbor cell type proportions could achieve comparable or better results than complex graph neural network models.[20] Motivated by these findings, we focused extensively on phenotypic screening using Cell Painting data in this study.”

• It seems to me that the study primarily focuses on datasets that are known to contain a high proportion of cytotoxicity-related assays. It would be beneficial to extend the analysis to datasets that include more diverse bioactivity endpoints. Does their conclusion also apply to other datasets with less cytotoxic compounds?

We have also now added a new curated benchmark (EvE Bio dataset)[29], which was just released and meets the goal of containing more diverse bioactivity endpoints. The EvE Bio benchmark (now described within Methods) comprises 24 curated protein target assays (agonist/antagonist) (Figure 11). On this benchmark, Cell Painting profiles outperform cell count, demonstrating the value of high-dimensional features for more specific endpoints, consistent with our overall findings. A detailed discussion of these results is now included within Results and Discussion.

We do hope that our paper draws attention to the fact that popular benchmarks do contain a high proportion of cytotoxicity-related assays; we are not convinced that this fact is well-known! We have added a discussion on this:

“A key limitation of all previously existing benchmark datasets from publicly available bioactivity data are that they include assays that are off-target or secondary pharmacology screens rather than solely measurements against the compounds’ intended primary targets. As a result, the compounds tested often share overlapping chemical features and target similar protein families, reducing the independence between assays and potentially inflating model performance due to correlated activity profiles. Further, the functional relevance of assay readouts can vary substantially depending on the compound’s mode of action. For example, antagonists tested in the absence of an agonist may not elicit functional responses detectable by high-content imaging, while agonists can induce receptor desensitization, influencing time-dependent phenotypes. These complexities limit the interpretability of morphological changes and challenge the ability to distinguish specific pharmacological mechanisms from indirect or off-target effects. Some compounds commonly annotated as GPCR actives may, in fact, be modulators of related pathways such as phosphodiesterases (e.g., IBMX), transporters, or metabolic enzymes.[54] Without well-controlled assay conditions, such compounds could be misclassified as receptor agonists or antagonists, further confounding model training and evaluation. Despite efforts to curate the dataset (such as removing assays with excessive cytotoxicity and cell line inhibition) these confounding factors remain inherent to the current public datasets.”

• Please discuss other potential biases beyond cytotoxicity (e.g., cell line effects, staining artifacts, variations in experimental conditions or protocols, etc.).

Thank you for the suggestion, this is a great idea. We have updated the Recommendations and added a discussion regarding using control wells and QC steps to mitigate such artifacts.

“Additional factors influencing model interpretation include the choice of cell lines, treatment durations, and compound concentrations.”

“Beyond cytotoxicity, additional sources of bias can influence predictive performance. These include cell line-specific responses, staining or imaging artifacts, and variability in experimental conditions or protocols (e.g., incubation times, compound dosing, or reagent preparation). While our cross-dataset analysis mitigates some of these issues, we recommend that studies explicitly evaluate such factors (e.g., following protocols like those of the OASIS[55]). In particular, chemicals should not be grouped onto plates by bioactivity; instead, randomized plate layouts are advisable. A key pitfall to avoid is running Cell Painting assays with the exact same plate layout as the target assay, because this allows technical artifacts to confound both readouts. Importantly, none of the datasets used in this study suffer from this issue.”

We have added new text discussing potential batch effects and confounders (Reviewer 3 raised this as well) - see the new section entitled “Consistent cell count reduction across plates shows benchmark active compounds often induce cell death”. We have included a new sample analysis of cell count at the per-plate level before aggregation, after normalising with respect to within-plate controls.

“In this study we leveraged Cell Painting data normalized against plate-specific DMSO controls and aggregated across replicates to mitigate technical variability inherent to high-throughput imaging assays. We also evaluated the cell count feature (among others) prior to aggregation and found that, although artifacts exist that can affect cell count, the predictive power of cell counts is not an artifact, but rather a real phenotypic effect. While the assay outputs we aim to predict may themselves be subject to technical or batch effects, it is important to note that these readouts originate from independent experiments distinct from the imaging data, with different plate layouts and batching, and without marked grouping of activities on particular plates.

Specifically, we found that of 402 plates from the Broad Cell Painting dataset, 99% contained both bioactive and inactive compounds, defined by activity annotations from the Hofmarcher, Ha, or Sanchez datasets. Notably, across these plates, bioactive compounds exhibited a consistent reduction in mean cell count relative to inactive compounds in 88% of cases, even before aggregating features at the compound level (Figure 7a). This reproducibility supports the relevance of cell count as a sensitive phenotypic marker.

We found that variability in pre-normalized cell count ranks moderately (94th out of 1783 features) in terms of mean standard deviation across plates. Features such as nuclei area shape exhibited even higher inter-plate variation (Figure 7b). Overall, the consistent decrease in cell count observed across nearly all plates for bioactive compounds reinforces confidence in cell count as a phenotypic marker; however, this also highlights that the benchmark assays currently used are heavily biased toward viability-related endpoints, and consequently, the identified bioactive compounds largely reflect viability effects.”

- **While the paper highlights the predictive power of cell count, it does not explore other potentially useful simple features, such as nuclear morphology or mitochondrial integrity. Are these also correlated with the endpoint?**

In our experience, many image-based features correlate with cell count, given that cell proliferation and cell crowding dramatically impact morphology, and we confirmed that for the Broad Cell Painting dataset - we have added a correlation plot of cell count with other features for bioactive vs inactive compounds:

“Next we aggregated the Cell Painting features as medians across replicates, and we filtered compounds to retain only unique compound skeletons in the bioactive and inactive groups, minimizing structural redundancy. Still, we found that bioactive compounds tend to have a larger number of Cell Painting features exhibiting higher absolute correlation with the cell count feature compared to inactive compounds (Figure 7c).

These findings suggest that bioactive compounds—likely those that reduce cell count—are associated with stronger phenotypic changes captured by the Cell Painting features. Conversely, inactive compounds display fewer such correlations, supporting the specificity of these features in reflecting relevant bioactivity impacting cell viability.”

Figure 8. (c) Distribution of the number of features with high absolute correlation to the cell count feature (after aggregation) for bioactive and inactive compounds (after chemical skeleton level deduplication).”

We have also explained our choice of cell count features and that the cell count feature contains a lot of the signal claimed from more complex features. This is why we used “MitoCount” from ScaleFEx as an example (Figure 8) and found that while it can predict toxicity at high concentrations, full profiles outperform it for subtle phenotypes. We now emphasize that the cell count is representative but not the only informative simple feature.

- It would be important if the authors provided a way to filter out datasets to remove associations highly correlated to cell count. Also, please subset existing datasets to remove this potential bias and provide this as a contribution to the community.

We have now released a filtered benchmark (available via Zenodo) and highlighted assays dominated by cell count (Figure 1, Supplementary Data). This is one of our main recommendations, and the code associated with the paper (see the Github link in the paper) demonstrates how the filtering is done, so that others can adapt to new datasets.

Although not directly related to this query, we also realized we should more carefully document recommendations for cleaning profiling datasets before using them in benchmark testing. We added to the recommendations a section on cleaning the Cell Painting datasets:

“For high-dimensional “profiling” datasets such as Cell Painting or gene expression, we recommend applying quality control (QC) post-processing, followed by plate- and batch-level corrections where necessary. Compounds that markedly reduce cell count—confirmed either visually or via cell count metrics—should be removed. To avoid confounding effects, features that are highly correlated with cell count (e.g., absolute correlation > 0.95) should also be excluded. Replicate perturbations can then be aggregated using the mean or median, but it is advisable to first address systematic artifacts, such as plate rows with consistently lower cell density.”

- The paper argues that cell count explains much of the predictive power, but it does not

fully dissect whether specific Cell Painting features provide additional mechanistic insights beyond viability.

We address this in two ways: anecdotally to show particular mechanistic insights, and more comprehensively. First, we evaluated the influence of individual features on the benchmark dataset from EveBio, which followed the recommendations of removing viability assays, promiscuous, and cytotoxic compounds.

“The assays most improved by including Cell Painting information were for GR agonism (mean AUC=0.79 with the complete Cell Painting profile vs AUC=0.60 with cell count baseline model) and RARb antagonism (AUC=0.83 vs AUC=0.64). The feature Nuclei_RadialDistribution_RadialCV_ER_4of4 was found to be the most predictive of this endpoint (Figure 11b); this feature is the coefficient of variation of intensity in the ER channel within the outermost region of the nucleus object. For RARb antagonism, the most predictive feature was Nuclei_Texture_Entropy_Mito_10_0 (Figure 11c), which measures the disorder or heterogeneity of the mitochondrial signal. Although mitochondria are cytoplasmic, limited image resolution and perinuclear overlap can cause mitochondrial signals to be captured within the nuclear mask. In both the GR agonist and RARβ antagonist cases, the phenotypes are subtle and not readily apparent by visual inspection (as confirmed in Supplementary Figures S4 and S5, which show no substantial changes in cell count). However, these effects are best captured through image-derived features. The superior performance of models using the full Cell Painting profiles compared to cell count baselines further supports the idea that spatial and texture-based features carry the relevant biological signal.”

We have also added a new section that is more comprehensive, comparing true predictions from the model using all Cell Painting features with true predictions from the model using the cell count feature:

“To assess whether the cell count baseline and the full Cell Painting feature model were identifying the same compounds as active, we examined the true positive prediction agreement between the two models across all assay endpoints in the Moshkov dataset. There was a 49.54% overlap between the true positive predictions, suggesting the approaches are partially predicting different groups of compounds as active. To investigate the difference in these compound cohorts, we isolated the compounds which were uniquely predicted by each model. We then calculated the following physicochemical properties for each compound molecule: molecular weight (MW), lipophilicity (LogP), topological polar surface area (TPSA), hydrogen bond donors (HBD) and acceptors (HBA) and rotatable bonds. We also computed the statistical significance of the difference of the means of each metric between the uniquely predicted compounds for each model using scipy’s “ttest_ind” ($p < 0.05$).

The true positive compounds predicted uniquely by the model trained only using cell count were found to have statistically significant lower average MW and LogP, while having a higher average TPSA and more rotatable bonds when compared with the cohort of compounds uniquely predicted by the model trained using all CellProfiler features (Supplementary Figure S2). Generally, lower LogP combined with higher TPSA indicates that the compound should exhibit better aqueous solubility.[44,45] Lower MW compounds also tend to diffuse more easily across cell membranes, and more rotatable bonds indicate higher molecular flexibility - more promiscuous binding and greater likelihood of fitting into various binding sites.[46]

Taken in combination this demonstrates that the cell count model is more adept at identifying compounds with more broad, simplistic effects and higher general/nonspecific cytotoxicity as they are more likely to impact broad cellular processes. Conversely, the model trained on all features is able to categorise compounds which have a lower likelihood of being reliably delivered to cells, or exhibit more subtle or specific phenotypes.

To further explore these findings, we examined the structural diversity of compounds that were correctly predicted as active by only one of the models. When comparing compound scaffolds, we observed substantially lower scaffold diversity among the compounds uniquely identified by the cell count model (528 unique scaffolds) compared with those uniquely identified by the full Cell Painting feature model (1,175 unique scaffolds). This further supports the finding that the cell count model is more narrow in its predictive ability, but when applied in a setting where a high number of compounds are cytotoxic, or assays are primarily aimed at measuring cell viability, this approach can appear to perform on par with a model utilizing all CellProfiler features, demonstrating the necessity for the benchmark curation suggested within this paper.”

Figure S2. Distribution of physicochemical properties for uniquely predicted true positive compounds from each model. Histograms compare the distributions of (A) molecular weight (MW), (B) lipophilicity (LogP), (C) topological polar surface area (TPSA), (D) number of hydrogen bond donors (HBD), (E) number of hydrogen bond acceptors (HBA), and (F) number of rotatable bonds between compounds uniquely predicted as true positives by the Cell Count Feature Model (cc_feat) and the All Feature Model (all_feat). Statistical comparisons were conducted using independent t-test; features with $p < 0.05$ are considered significantly different.

Further our analysis on the benchmark datasets and Figure 11 shows that Cell Painting features do indeed capture assays beyond viability. We have now expanded the discussion on important Cell Painting features on a case-by-case basis.

“Overall, the value of imaging-derived features largely hinges on the screening context. Previous studies have explored the importance of particular Cell Painting features with respect to mitochondrial toxicity (edge of the mitochondria object)[22] and cytotoxicity prediction in muscle cells (spots edges and ridges profiles in the ER and mitochondria channels)[53]. Our results underscore the utility of the complete Cell Painting profile as high-content imaging data for capturing biologically meaningful phenotypes associated with signaling pathways, not just limited to cell count. These phenotypic responses become more detectable when cytotoxicity is not pervasive in the dataset, causing models to be focused on subtle cellular changes rather than cell count.”

- **The manuscript’s analysis of concentration-dependent effects is compelling but restricted to only six compounds. This small sample size undermines the generalizability of the findings and the claim that multi-dose experiments are critical for robust phenotypic profiling.**

After the paper was sent for review, a subset of our authors pre-printed a study on a larger set of appropriate data for this analysis, over 1,000 compounds - at our request, they included cell count analysis in their study (Ewald et al. <https://pmc.ncbi.nlm.nih.gov/articles/PMC11785178/>). We now refer to this work in our discussion, which supports our findings that morphology changes precede cell death.

“Although these findings are preliminary and specific to the six compounds, this is consistent with our larger analysis in Ewald et al., which found that across >1000 compounds, early morphology changes were observed at doses about an order of magnitude lower than cytotoxicity readouts. Further studies affirm that for most compounds that exhibit morphological changes, the information content in Cell Painting is dependent on dose. [Ewald et al 2025]”

- **There is an unclear breakdown of assay types. The authors note that many benchmark assays are related to cell viability or cytotoxicity, but they do not quantify or categorize the assay types across datasets. Could they provide a table that breaks down the assay type in each dataset? (e.g., viability, protein-target, etc.)**

Good idea. We now include a table (Supplementary Table S1) and figure (Supplementary Figure S1) categorizing assays based on ChEMBL categorization: e.g., viability (cell-line) vs. protein target (single protein). This was only possible for public datasets where such information was available. Not all assays from all sources (particularly the Broad dataset) were annotated to make this classification possible.

- The manuscript employs different baseline models (e.g., logistic regression, XGBoost) across analyses, which may be confusing about the primary baseline being advocated and complicate comparisons. XGBoost with a single feature is an overkill. Can the authors also use the raw cell count as a prediction score to understand its predictive power?

We now clarify that the raw cell count feature was used directly in threshold-based predictions (Figure 1). We used LR for consistency and interpretability, and XGBoost only when required for direct comparison to prior benchmarks. We added this clarification to the Results and Methods.

“We first evaluated baseline cell count models trained on a single image-based feature, namely cell count.... This allows us to systematically assess the predictive utility of cell count as a prediction score to understand its predictive power.”

“We systematically assessed the predictive utility of the cell count feature across multiple assays by using the raw cell count as a prediction score to understand its predictive power by evaluating classification performance at different thresholds.”

- Why did the authors use only molecular weight and $\log P$ as chemical features? What is the rationale behind choosing only these?

These were chosen to match prior CLOOME benchmarks which had access to the full chemical structure. Molecular weight and $\log P$ were chosen as proxies, as these are known confounding factors in the chemical assays. We now clarify this and also show that adding them to the cell count improves performance slightly (Table 2, row d).

“Since CLOOME embeddings have access to chemical structures, two baseline cell count models were considered: (1) using cell count as the sole feature and (2) using cell count along with molecular weight and $\log P$ (calculated using RDKit) as very simple chemical features. The first baseline excludes any chemical information, while the second baseline, although not explicitly trained on chemical features, incorporates some chemical-related information indirectly through molecular weight and $\log P$. Molecular weight and $\log P$ were

chosen as proxies for chemical information, as these are known confounding factors in the chemical assays.[25,26]"

Reviewer #2 (Remarks to the Author):

This is an interesting and valuable study that challenges prevailing assumptions in phenotypic screening by showing that simple cell count features can perform comparably to more complex morphological or transcriptomic features (e.g., Cell Painting or gene expression) in predicting compound bioactivity. The manuscript is well-written and addresses a timely and relevant question in the field of machine learning for drug discovery. I appreciate the authors' efforts in systematically reevaluating benchmark datasets and proposing practical recommendations. However, I believe the manuscript would benefit from addressing the following points before publication:

1. On the specificity of cell count as a proxy for toxicity:

The authors argue that many assays are primarily capturing cell health or cytotoxicity signals, which may explain why cell count alone performs well. However, cytotoxicity is not binary—it can vary in intensity and specificity. Could cell count only detect strong cytotoxic effects while missing more nuanced or phenotype-specific signals? I encourage the authors to further clarify the scope and limits of what cell count can capture, perhaps by stratifying compounds by degree or type of toxicity and evaluating the cell count model's sensitivity.

Great point - we now add a new analysis on this. Figure 1g now shows the cell count feature value for compounds that are active in n assays (combining Moshkov and Hofmarcher datasets). This shows that cell count contains information that is increasingly more predictive of strong and promiscuous cytotoxic responses across different cell types, and less for subtle phenotypes.

We have added a new discussion to this effect:

“To better contextualize the scope of what cell count captures, we analyzed whether the predictive signal from cell count correlates with cytotoxicity. Specifically, we combined the Moshkov and Hofmarcher datasets and stratified compounds by the number of assays in which they were labeled cytotoxic. We compared the cell count values for compounds active in multiple assays and found that compounds active in a greater number of these benchmark assays tend to show a stronger and more consistent decrease in cell count (Figure 1g). Overall, this indicates that the cell count feature is more predictive for compounds that are promiscuous.”

2. On the complementarity between deep learning models and cell count models: While the authors compare cell count with several state-of-the-art models, the current manuscript does not analyze what differences exist between the compounds identified as active by different methods. A Venn diagram is not enough. I suggest the authors perform a deeper analysis of the compounds that are uniquely identified by deep learning models but missed by cell count models. What are the chemical or functional characteristics of these compounds? It is possible that these uniquely identified compounds have more subtle or complex modes of action, and understanding this may significantly strengthen the paper's conclusions.

This is an interesting exploratory direction! We performed two new analyses related to this: one looking for mechanistic insights at the assay level (which assays are better-predicted with full Cell Painting profiles vs cell count) - see above (Reviewer 1 comment "The paper argues that cell count explains much of the predictive power, but it does not fully dissect whether specific Cell Painting features provide additional mechanistic insights beyond viability")

At the compound level, for the Moshkov dataset only, the authors released predictions from each compound in the original study, and we used these predictions to conduct a new analysis (now included within the paper in section "Appropriate evaluation metrics and assay endpoints are essential when modelling large-scale benchmark datasets"):

"To assess whether the cell count baseline and the full Cell Painting feature model were identifying the same compounds as active, we examined the true positive prediction agreement between the two models across all assay endpoints in the Moshkov dataset. There was a 49.54% overlap between the true positive predictions, suggesting the approaches are partially predicting different groups of compounds as active. To investigate the difference in these compound cohorts, we isolated the compounds which were uniquely predicted by each model. We then calculated the following physicochemical properties for each compound molecule: molecular weight (MW), lipophilicity (LogP), topological polar surface area (TPSA), hydrogen bond donors (HBD) and acceptors (HBA) and rotatable bonds. We also computed the statistical significance of the difference of the means of each metric between the uniquely predicted compounds for each model using scipy's "ttest_ind" ($p < 0.05$).

The true positive compounds predicted uniquely by the model trained only using cell count were found to have statistically significant lower average MW and LogP, while having a higher average TPSA and more rotatable bonds when compared with the cohort of compounds uniquely predicted by the model trained using all CellProfiler features (Supplementary Figure S2). Generally, lower LogP combined with higher TPSA indicates that the compound should exhibit better aqueous solubility.[44,45] Lower MW compounds also tend to diffuse more easily across cell membranes, and more rotatable bonds indicate higher molecular flexibility - more promiscuous binding and greater likelihood of fitting into various binding sites.[46] Taken in combination this demonstrates that the cell count model is more adept at identifying compounds with more broad, simplistic effects and higher general/nonspecific cytotoxicity as they are more likely to impact broad cellular processes. Conversely, the model trained on all features is able to categorise compounds which have a lower likelihood of being reliably delivered to cells, or exhibit more subtle or specific phenotypes.

To further explore these findings, we examined the structural diversity of compounds that were correctly predicted as active by only one of the models. When comparing compound scaffolds, we observed substantially lower scaffold diversity among the compounds uniquely identified by the cell count model (528 unique scaffolds) compared with those uniquely identified by the full Cell Painting feature model (1,175 unique scaffolds). This further supports the finding that the cell count model is more narrow in its predictive ability, but when applied in a setting where a high number of compounds are cytotoxic, or assays are primarily aimed at measuring cell viability, this approach can appear to perform on par with a model utilizing all CellProfiler features, demonstrating the necessity for the benchmark curation suggested within this paper.”

Figure S2. Distribution of physicochemical properties for uniquely predicted true positive compounds from each model. Histograms compare the distributions of (A) molecular weight

(MW), (B) lipophilicity (LogP), (C) topological polar surface area (TPSA), (D) number of hydrogen bond donors (HBD), (E) number of hydrogen bond acceptors (HBA), and (F) number of rotatable bonds between compounds uniquely predicted as true positives by the Cell Count Feature Model (cc_feat) and the All Feature Model (all_feat). Statistical comparisons were conducted using independent t-test; features with $p < 0.05$ are considered significantly different.

3. On clarity of the conclusions:

Although the authors have conducted extensive experiments, the conclusions, particularly in the discussion section, are not sufficiently distilled. I suggest the authors revise the discussion to clearly articulate their key findings, implications for future benchmarking practices, and the broader impact of their work on method development in the field.

We appreciated this feedback and rewrote the conclusions to clearly outline key takeaways and our five recommendations for future benchmarking.

4. On model simplicity and interpretability:

The authors make an important point regarding the value of simple and interpretable methods in bioactivity prediction. This aligns with recent findings in other domains of computational biology. For example, SpaDo (Genome Biology, 2024) showed that in spatial transcriptomics, simple clustering based on spatial neighbor cell type proportions could achieve comparable or better results than complex graph neural network models. I recommend that the authors cite such work to further reinforce the message that biological interpretability and simplicity should be weighted more heavily in computational model development.

We now cite SpaDo (Genome Biology 2024) and other examples emphasizing model simplicity in biology. Thank you for this valuable suggestion. We have added a discussion on the same.

“Previous studies have benchmarked models using gene expression data[18]. For example, Csendes et al. found that simple baselines outperformed state-of-the-art foundation models in predicting post-perturbation gene expression.[19] Similarly, SpaDo showed that in spatial transcriptomics, simple clustering based on spatial neighbor cell type proportions could achieve comparable or better results than complex graph neural network models.[20] Motivated by these findings, we focused extensively on phenotypic screening using Cell Painting data in this study.”

Reviewer #3 (Remarks to the Author):

Summary

This manuscript presents an analysis of assay predictability from image-based phenotypic assays, specifically the Cell Painting assay, and the role of cell count as a feature that can drive performance. The problem of assay prediction is very complex due to the multiple data sources that interact and the many confounding factors that can lead to incorrect interpretations. The analysis and effort presented by the authors are a valuable step forward to improve clarity about how this problem can be approached and what corrections need to be made going forward to advance the state of the art in this field.

While the questions and presented analysis are promising, the main limitation of this study is that it continues to rely primarily on the same phenotypic dataset, which may no longer be useful to make progress in assay prediction problems. The study makes the assumption that cell count is a clean readout, while in practice, the dataset used may have significant technical artifacts due to plate layouts and batch effects. The pattern reported in this paper may be a more fundamental issue with the imaging dataset instead of being a generalized phenotypic property for assay prediction. In a related note, the assay endpoints across the three benchmark sets analyzed may all be related to each other because they may activate the same (polypharmacological) compounds. These problems are part of the challenges of the assay prediction problem and need to be addressed in this study to more convincingly support the conclusions.

Main comments

1. and 2. Figure 1 shows cell count distributions, and Figure 2 shows how many assays can be predicted with cell counts relative to other predictors. The main question is: are cell counts batch-effect independent? Or are cell counts biased by batch effects as much as other features? It is well known that cell density is a significant source of technical (not phenotypic) variation, and the cpg0012-wawer-bioactivecompoundprofiling dataset may have significant batch biases, because bioactive compounds were placed in certain plates different from those used for non-bioactive compounds. How are cell counts correlated with plates or batches? A potential answer could be obtained by looking at the distribution of cell counts in the control well only, but other types of analysis may be needed to clarify this question.

If artifacts that affect cell count exist, how can we make sure that the predictive power of cell counts is not an artifact, but rather a real phenotypic effect? The main concern is that all the predictors (including cell counts) may be leveraging technical variation to predict the same subset of bioactive compounds (Fig. 2), which may come from the same plates (i.e., same technical artifact).

The reviewer raises a very important point here, which we have addressed in several ways in the revised manuscript. First, we should have mentioned in the paper that the data used in modelling used controls to normalize features, including cell counts, on a per-plate basis; this should mitigate (though may not eliminate) plate-level and batch-level effects. We have now added a statement to this effect in the methods:

“Thus, the features from the Cell Painting dataset used in this study are the deviation from DMSO controls on the same plate after normalization, and after aggregating median across replicates.”

That said, we began to address the concern by analyzing the placement of bioactive vs inactive compounds in the Cell Painting data. Among the 402 plates analyzed, 398 contained both bioactive and inactive compounds (defined as being active in at least one assay in Hofmarcher, Ha or Sanchez datasets). Of these 398 plates, bioactive compounds exhibited a lower mean cell count than inactive compounds in 351 plates. The remaining 4 plates, which contained only bioactive compounds, included just 8 compounds in total. We added a discussion of this and emphasize why our data uses plate negative controls (DMSO) for normalization:

Figure 7. (a) Distribution of the difference between mean cell count values for bioactive versus inactive compounds across 402 plates from the Broad Cell Painting dataset. (b) Ranking of top 50 Cell Painting features by mean standard deviation across plates; cell count (not shown) ranks 94th out of 1783 features in inter-plate variability. (c) Distribution of the number of features with

high absolute correlation to the cell count feature (after aggregation) for bioactive and inactive compounds (after chemical skeleton level deduplication)

“Consistent cell count reduction across plates shows benchmark active compounds often induce cell death

Cell count can be impacted by many technical variables, raising the potential for plate or batch effects to confound analysis. “In this study we leveraged Cell Painting data normalized against plate-specific DMSO controls and aggregated across replicates to mitigate technical variability inherent to high-throughput imaging assays. We also evaluated the cell count feature (among others) prior to aggregation and found that, although artifacts exist that can affect cell count, the predictive power of cell counts is not an artifact, but rather a real phenotypic effect. While the assay outputs we aim to predict may themselves be subject to technical or batch effects, it is important to note that these readouts originate from independent experiments distinct from the imaging data, with different plate layouts and batching, and without marked grouping of activities on particular plates.

Specifically, we found that of 402 plates from the Broad Cell Painting dataset, 99% contained both bioactive and inactive compounds, defined by activity annotations from the Hofmarcher, Ha, or Sanchez datasets. Notably, across these plates, bioactive compounds exhibited a consistent reduction in mean cell count relative to inactive compounds in 88% of cases, even before aggregating features at the compound level (Figure 7a). This reproducibility supports the relevance of cell count as a sensitive phenotypic marker.

We found that variability in pre-normalized cell count ranks moderately (94th out of 1783 features) in terms of mean standard deviation across plates. Features such as nuclei area shape exhibited even higher inter-plate variation (Figure 7b). Overall, the consistent decrease in cell count observed across nearly all plates for bioactive compounds reinforces confidence in cell count as a phenotypic marker; however, this also highlights that the benchmark assays currently used are heavily biased toward viability-related endpoints, and consequently, the identified bioactive compounds largely reflect viability effects.”

Next we aggregated the Cell Painting features as medians across replicates, and we filtered compounds to retain only unique compound skeletons in the bioactive and inactive groups, minimizing structural redundancy. Still, we found that bioactive compounds tend to have a larger number of Cell Painting features exhibiting higher absolute correlation with the cell count feature compared to inactive compounds (Figure 7c).

These findings suggest that bioactive compounds—likely those that reduce cell count—are associated with diverse phenotypic changes captured by the Cell Painting features. Overall, we found that the predictive power of cell counts is not an artifact of actives being concentrated on particular plates in the experiment, but rather a real phenotypic effect. ”

We agree that prediction of assay activity may in theory be confounded by plate layout effects; however, in this study we used three different sources of Cell Painting datasets – AstraZeneca

dataset, Broad dataset, and the NY Stem Cell Foundation dataset – which all had differing plate and batch layouts for compounds making it impossible for those to be predictive for assay endpoints which came from still other plate and batch layouts. The assay outputs that we predict may also have technical/batch effects, but they are completely separate experiments, which makes it more unlikely that batches or plates or well positions ‘match’ the imaging data.

The assay outputs we aim to predict may also contain technical or batch effects; however, because each assay output experiment is an entirely separate experiment from the Cell Painting experiment (different laboratory, typically testing different chemical libraries in different plate/batch layouts), it is unlikely that batch, plate, or well-position effects align with the imaging data in any consistent way.

Our results are overall the same across the datasets, that is, cell count is a good predictor of “activity”. Because cell death is essentially represented in cell counts, this seemed a reasonable hypothesis, as “active” compounds mostly have lower cell count compared to inactive compounds (across all three Cell Painting datasets). We show that drop in cell count for “active” compounds is consistent across plates where both inactive and bioactive compounds are present.

We have now added a new section on the “Limitations of this Study”

“We also acknowledge that technical artifacts, such as plate layout effects, can confound assay activity predictions, particularly when bioactive compounds cluster on specific plates. In the case of the Broad dataset, by analyzing plate-level data, we demonstrate consistent reductions in cell count for bioactive compounds (as defined by current benchmarks) across plates, indicating that the predictive power of cell count is unlikely driven solely by technical bias. Furthermore, given that the Broad, AstraZeneca, and New York Stem Cell Foundation Cell Painting datasets image sets were all independently created and not likely to follow the same plate and batch layouts, this rules out artifacts as yielding falsely positive results; for new datasets, the potential impact of plate and batch layout needs to be evaluated for each case separately.”

3. Fig. 1 shows similar results in three datasets, but in reality, it’s the same phenotypic dataset. In other words, the cell counts come from the same imaging dataset. The difference between the three datasets is the assays only. In fact, the Hofmarcher and Ha datasets appear to be almost identical except for a handful of assays, according to the text and references. Is there any relationship between the Moshkov and Hofmarcher assays? Are the compounds labeled as active in Moshkov also active in Hofmarcher? Ultimately, it would be best to unify the set of assays into a single coherent benchmark. The presentation of results in three datasets (Moshkov, Hofmarcher, and Ha) is artificial, and makes it look as if the problem is widespread, but ultimately, we are talking about the same phenotypic dataset. A unified view should be presented, with similar technical choices, such as cross-validation differences, and so on.

We now clarify in the text that the Moshkov dataset (assays performed internally at the Broad Institute) is distinct from the Hofmarcher/Ha datasets, which are indeed derivatives of each other, both being collected from ChEMBL, but collected at different times, resulting in a few different

compounds and assays. We now emphasize that while the assays differ to some extent, the imaging data used to predict them is shared across the Hofmarcher/Moshkov/Ha benchmarks. We also explain the decision not to group all assays into a single dataset so that we could directly compare our results against prior published results (e.g., Figures 1–2). We have now added text to Results to this effect:

“The Moshkov dataset, containing assay readouts generated internally at the Broad Institute, is distinct from the Hofmarcher and Ha datasets, both derived from ChEMBL. Sanchez-Fernandez et al. based their benchmarking on the Hofmarcher dataset. Although Hofmarcher and Ha originate from the same source, they were compiled at different times, leading to minor differences in compound coverage. All three datasets were evaluated against the same Cell Painting imaging dataset. To allow direct comparison with prior studies, we evaluated each dataset separately rather than combining them into a single benchmark.”

We now also provide a new benchmark dataset from EveBio that is a different source for assays - it measures the agonism and antagonism of compounds across various targets, with compounds that induce viability removed from both the target assay space and the Cell Painting space; hence models here learn activity free from large deviations in cell count. We quote part of the new section below:

“In order to facilitate the above recommendations, we are together with this work releasing a benchmark dataset for compound-activity prediction curated along the recommendations above from EvEBio.[52] This benchmark dataset comprises ligand binding and functional assays in biochemical systems for nuclear receptors (NRs), alongside cell-based activity measurements for seven-transmembrane receptors (7TMs), supplemented by cell viability assays to monitor cytotoxicity, collectively covering a broad panel of these receptor classes. After applying the same filtering steps described above (see Benchmarking Cell Painting for curated Protein-Target data) we isolated a high-quality benchmark of 24 protein-target assays (including agonist and antagonist modes) with binary activity labels (active/inactive) across 269 compounds which also have Cell Painting data publicly available in the Broad dataset. Using this benchmark, models leveraging the full Cell Painting profiles outperformed the baseline cell count model, achieving a higher mean balanced accuracy (BA = 0.67 vs. 0.52; see Figure 11a). Compared to both a randomly shuffled baseline and the cell count baseline, the full-profile models delivered over 20% improvement in AUC-PR on 8 of the 24 tasks (Supplementary Figure S3). Notably, gains in AUC-ROC were consistent regardless of the test set size or the proportion of active compounds, suggesting that these improvements are robust and not driven by dataset composition (Supplementary Figure S3).”

4. To confirm the claim more broadly, another phenotypic dataset should be analyzed as well. The manuscript mentions the JUMP dataset, for instance, which could be used as an independent source of phenotypic variation to investigate the role of cell counts for assay prediction. If the same compounds are present across the dataset, these can help to further investigate if the patterns observed in Fig. 1 and others are prevalent beyond the dataset that the community has used. Plate layouts will be different, and technical variation may have different biases, which can lead to different results. The paper presents preliminary results with another set in Figure 8, but its scope is limited to supporting the broader claims.

The JUMP-CP dataset is not yet fully prepared for batch-corrected analysis and was therefore excluded from this study. Instead, in the revised manuscript we present results from two additional Cell Painting chemical-perturbation datasets beyond the Broad dataset: the New York Stem Cell Foundation dataset (Comolet et al.), and the AstraZeneca dataset (Cross-Zamirski et al.). We show that models trained to predict targets, compound dose, or positive controls (the task differs depending on the dataset) perform comparably to cell count–based models.

Across all datasets, the conclusions remain consistent: bioactive compounds tend to show a reduction in cell count relative to inactive compounds in most assays in current popular benchmarks. When benchmark tasks are biased toward viability-related endpoints, cell count alone is often sufficient to predict assay outcomes from imaging data.

We have now added a statement to this effect in the Limitations Section:

“Across all datasets analyzed—the Broad, AstraZeneca, and New York Stem Cell Foundation Cell Painting datasets—we evaluated whether cell count alone could predict various assay outcomes or intended tasks set in the original studies (such as separating drugs from positive controls). In each dataset, we found that cell count performed comparably to models using full Cell Painting profiles. Our results consistently show that bioactive compounds, as defined in these benchmarks, tend to reduce cell count, especially when the assays in these benchmarks are biased toward viability-related endpoints.”

5. Continuing with the batch effects issue in the selected dataset, are the models labeled as “All Features – Logistic Regression” in Figures 5 and 6 batch corrected? While the architecture is simpler, the features may be contaminated by technical variation that reveals where the samples come from (bioactive plates) rather than finding real morphological signals.

Yes. We have now clarified that CellProfiler features were plate-normalized and DMSO-corrected as in prior work. The reviewer is correct that in the original Bray 30k compound set, the “bioactive” set of compounds was indeed a separate batch of plates, so they were quite concentrated. Here, however, we are using bioactives to mean the ones annotated as active in the assay dataset, which does not relate to what Bray et al refer to as bioactive. We also showed that, as per assay activity, both bioactive and inactive compounds (by our definition) are present in each plate of the Cell Painting dataset from the Broad Institute rather than concentrated on a few plates; further, to model, we used median aggregated profiles, which further reduces confounding technical factors as replicates are spread over separate plates (though typically in the same well position within each plate). We have added this to the Methods and Results sections as indicated above.

6. In a separate note, one of the recommendations for benchmarking is to keep assays that have sufficient positive and negative examples (more than 40). However, in many assay prediction problems, collecting such a high number of positive examples is unrealistic or simply defeats the purpose of predictive modeling because 40 hits may already be sufficient to move a drug discovery project forward. While having a high number of positive and negative examples is desirable for machine learning researchers, how can the authors include realistic, rare hit assays in their benchmarks? It is highly

recommended to consider such examples, which may be more interesting for the community in practice.

This point is well-taken; we have added this point to our recommendations. We suggest using balanced accuracy or qualitative analysis for assays with few positive example compounds and emphasize that prediction may still be useful with few actives. Apart from the ‘project’ view, we also emphasize there needs to be more focus on understanding readouts and technologies better. We have now reworded the recommendations to:

“For example, with models that achieve an AUC of 0.85, we recommend using AUC as a metric only when the held-out test sets contain at least 20 active and inactive compounds.⁴² When this is not feasible (for assays with few positive example compounds), AUC may not be an appropriate metric. Instead, we suggest evaluating models using absolute predictions instead of predicted probabilities, using metrics such as balanced accuracy, while carefully accounting for potential confounding effects from data imbalance using random prediction baselines and y-scrambling.”

Other comments:

7. Figure 1(a-c): Are the points compounds or assays? If compounds, it seems too few points with respect to what’s reported in the text (10K to 16K). If assays, it seems too many points with respect to what the caption says (~200). How were these points selected for the plot?

The points represent compounds; given there are too many to represent, the Seaborn swarmplot overlaps and clusters them together, especially in areas of high density. We have now addressed this in the Figure.

8. Are the phenotypes presented in Figure 4 consistent across replicates?

Yes, these were consistent across all sites and replicates; for the reader we used a single image. We have now linked the visualizations to the data viewer portal that shows images from different sites and replicates. "Images of cells from other sites and replicates of these compounds show the same trends and are viewable at <https://idr.openmicroscopy.org>"

9. Lines 203 to 212 assert that AUPRG has benefits over AUPRC and AUC, giving the impression that the metrics were going to be unified and reported with the suggested metric. However, the results in the main manuscript are all based on AUC-ROC only. Why is the AUC-ROC adopted in this manuscript if another metric may have better interpretation properties?

We were limited to AUC-ROC to enable direct comparison to prior published results, but used AUPRG for a comparative answer. We do compare results with AUCPR in certain cases where necessary to understand the predictivity of bioactive compounds. We have clarified the language in the text to reflect the same.

"AUPRG provides a universal baseline, which is comparable across assays with different class distributions (unlike Area Under the Precision-Recall Curve), and is less impacted than AUC by the abundance of negative class examples that are present in these classification tasks. For other tasks, we limited the analysis AUC-ROC to enable direct comparison to prior published results, but used AUPRG and AUCPR for a comparative answer."

Reviewer #1 (Remarks to the Author):

Thank you for carefully addressing all the comments and suggestions from my previous review. I have gone through the revised version of your manuscript, and I am pleased to see that my concerns have been resolved. The revisions have strengthened the paper and improved its overall clarity and impact.

I have no further comments and recommend the manuscript for publication.

Reviewer #1 (Remarks on code availability):

I checked the repository but did not run the code.

Reviewer #2 (Remarks to the Author):

The manuscript has been substantially improved, and most previous concerns have been fully addressed. The additional analyses greatly enhance the clarity and rigor of the work.

One minor issue remains: while Figure 1g clarifies that cell count captures strong and promiscuous cytotoxic responses, the manuscript still does not explicitly discuss its limited sensitivity to subtle or mechanism-specific effects. I suggest adding a brief statement in the Discussion to acknowledge this limitation.

Response:

We have modified the sentence in the “Multiple bioactivity tasks are highly correlated with the simple cell count feature” section in response:

“While cell count is informative for strong and promiscuous cytotoxic responses, it is considerably less responsive to mechanism-specific or low-grade phenotypic changes, which may constrain its utility for detecting nuanced biological effects.”

With this minor revision, the paper will be suitable for publication.

Reviewer #4 (Remarks to the Author):

Authors have performed a comprehensive and timely benchmark study about cell counting and Cell Painting verified from multiple aspects. The manuscript is well-organized and the results are comprehensive and convinced. The editor contacted me and asked me to review the author's responses to Reviewer #3's concerns. Authors have completely addressed the concerns proposed by Reviewer #3. The revised manuscript is well-written and the response letter is detailed. I have no additional suggest